# Aberrant DNA Methylation, Expression, and Occurrence of Transcript Variants of the ABC Transporter *ABCA7* in Breast Cancer

**DOI:** 10.3390/cells12111462

**Published:** 2023-05-24

**Authors:** Katja Zappe, Antonio Kopic, Alexandra Scheichel, Ann-Katrin Schier, Lukas Emanuel Schmidt, Yasmin Borutzki, Heidi Miedl, Martin Schreiber, Theresa Mendrina, Christine Pirker, Georg Pfeiler, Stefan Hacker, Werner Haslik, Dietmar Pils, Andrea Bileck, Christopher Gerner, Samuel Meier-Menches, Petra Heffeter, Margit Cichna-Markl

**Affiliations:** 1Department of Analytical Chemistry, Faculty of Chemistry, University of Vienna, 1090 Vienna, Austria; 2Department of Inorganic Chemistry, Faculty of Chemistry, University of Vienna, 1090 Vienna, Austria; 3Department of Obstetrics and Gynecology and Comprehensive Cancer Center, Medical University of Vienna, 1090 Vienna, Austria; 4Center for Cancer Research and Comprehensive Cancer Center, Medical University of Vienna, 1090 Vienna, Austria; 5Division of Gynecology and Gynecological Oncology, Department of Obstetrics and Gynecology, Medical University of Vienna, 1090 Vienna, Austria; 6Department of Plastic and Reconstructive Surgery, Medical University of Vienna, 1090 Vienna, Austria; 7Division of Visceral Surgery, Department of General Surgery and Comprehensive Cancer Center, Medical University of Vienna, 1090 Vienna, Austria; 8Joint Metabolome Facility, University of Vienna and Medical University of Vienna, 1090 Vienna, Austria

**Keywords:** ABCA7, ABC transporter, breast cancer, DNA methylation, gene expression, alternative splicing, intron retention/altered intron termination, mass spectrometry-based shotgun proteomics, doxorubicin, paclitaxel

## Abstract

The ABC transporter ABCA7 has been found to be aberrantly expressed in a variety of cancer types, including breast cancer. We searched for specific epigenetic and genetic alterations and alternative splicing variants of ABCA7 in breast cancer and investigated whether these alterations are associated with ABCA7 expression. By analyzing tumor tissues from breast cancer patients, we found CpGs at the exon 5–intron 5 boundary aberrantly methylated in a molecular subtype-specific manner. The detection of altered DNA methylation in tumor-adjacent tissues suggests epigenetic field cancerization. In breast cancer cell lines, DNA methylation levels of CpGs in promoter-exon 1, intron 1, and at the exon 5–intron 5 boundary were not correlated with *ABCA7* mRNA levels. By qPCR involving intron-specific and intron-flanking primers, we identified intron-containing *ABCA7* mRNA transcripts. The occurrence of intron-containing transcripts was neither molecular subtype-specific nor directly correlated with DNA methylation at the respective exon–intron boundaries. Treatment of breast cancer cell lines MCF-7, BT-474, SK-BR3, and MDA-MB-231 with doxorubicin or paclitaxel for 72 h resulted in altered *ABCA7* intron levels. Shotgun proteomics revealed that an increase in intron-containing transcripts was associated with significant dysregulation of splicing factors linked to alternative splicing.

## 1. Introduction

ATP-binding cassette (ABC) transporters are large, evolutionary highly conserved integral membrane proteins. By using the energy of ATP hydrolysis, they can transport exogenous and endogenous substances across membranes, even against steep concentration gradients [1]. ABC transporters are composed of four core domains: two transmembrane domains (TMDs) that are embedded in the membrane bilayer and determine ligand specificity, and two ATP-binding domains located in the cytoplasm [2]. Based on sequence homology and structural organization, the human ABC transporters known so far have been divided into seven subfamilies (ABCA–ABCG) [3]. The number of human ABC transporters reported ranges from 48 to 51, depending on whether pseudogenes were taken into account or not.

ABCA7 was identified by Kaminski et al. in 2000. They detected high *ABCA7* mRNA levels in peripheral leukocytes, thymus, spleen, and bone marrow, as well as lower levels in pancreas, liver, and lung tissue [4]. *ABCA7* expression has also been reported for microglia and neurons in the human brain [5] as well as white adipose tissue in mice [6]. The human *ABCA7* gene is located on chromosome 19p13.3, with the upper strand being the coding strand, and contains 47 exons. ABCA7 has 54% sequence identity with ABCA1 [4], which mediates the transport of cellular phospholipids and free cholesterol to extracellular apolipoproteins, in particular apolipoprotein A-I (apoA-I) [7]. There is increasing evidence that ABCA7 is also involved in lipid transport processes and cholesterol homeostasis [4,8,9,10,11,12]. Recent results from cell lipid efflux assays and shotgun lipid mass spectrometry suggest that ABCA7 has lower specificity for phosphatidylcholine, similar preference for cholesterol, and higher preference for non-choline-containing phospholipids compared to ABCA1 [13]. Several studies hint at a function of ABCA7, but not of ABCA1, in phagocytosis of apoptotic cells [9,14,15].

Genome-wide association studies (GWAS) identified *ABCA7* as one of the key risk genes of Alzheimer’s disease (AD), the most common cause of dementia. Loss of function of ABCA7 is assumed to contribute to AD-related phenotypes by enhancing cleavage of amyloid precursor protein (APP), resulting in amyloid-β accumulation in extracellular plaques in the brain [16]. Knockdown studies suggest that ABCA7 loss of function not only accelerates the formation of amyloid-β but also impairs its removal [17]. Loss of function of ABCA7 has been found to be associated with alterations in the DNA methylation status of cytosine guanine dinucleotides (CpGs) in or close to *ABCA7* [18,19,20], single-nucleotide polymorphisms (SNPs) in exons [21] or introns [22,23], and alternative splicing events [24].

To date, dysregulation of ABCA7 in cancer has been reported in only a few studies. By investigating transcript levels of human ABC transporters in pancreatic ductal adenocarcinoma, *ABCA7* was one of 20 ABC transporters found to be significantly up-regulated in tumor tissues compared to adjacent non-neoplastic tissues [25]. In a study on epithelial ovarian cancer, intraperitoneal metastases showed significantly higher *ABCA7* mRNA levels than primary tumors and control ovarian tissues [26]. Liu et al. found high *ABCA7* mRNA levels in ovarian cancer tissues to be associated with poor overall survival [27]. In the same study, down-regulation of *ABCA7* in the ovarian cancer cell line SKOV-3 decreased cell migration, increased E-cadherin levels, and decreased N-cadherin levels, suggesting a role of ABCA7 in epithelial–mesenchymal transition (EMT) [27]. Post-transcriptional silencing of *ABCA7* by microRNA 197-3p (miR-197) decreased expression of mesenchymal markers, including N-cadherin, vimentin, and snail proteins [28]. In a study on colorectal cancer, patients with low *ABCA7* mRNA levels had shorter disease-free interval after adjuvant chemotherapy than those with high *ABCA7* mRNA levels [29]. By analyzing expression levels of ABC transporters in breast cancer patients treated with neoadjuvant chemotherapy (5-fluorouracil-, doxorubicin-, cyclophosphamide- and/or paclitaxel-based), *ABCA7* was one of 22 ABC transporters being up-regulated in tumor tissues compared to paired non-neoplastic tissues [30]. All these findings suggest an association of ABCA7 with various types of cancer. However, the role and underlying mechanism of ABCA7 in cancer are yet to be discovered.

In the present study, we investigated if specific epigenetic and genetic alterations and transcript variants of *ABCA7* occur in breast cancer, and if detected alterations are associated with ABCA7 expression. We focused on breast cancer due to the accumulating evidence that dysregulation of cholesterol homeostasis is linked to breast cancer pathophysiology [31,32,33,34]. Increased levels of cholesterol and other serum lipids have been associated with obesity and a high risk of breast cancer development and progression [33,35]. In addition, growing evidence suggests that adipocytes, reported to express ABCA7 [6], have a promoting effect on breast cancer cells, presumably by secretion of a variety of molecules including inflammatory cytokines, hormones, adipokines, and lipid metabolites [36,37,38,39].

Our study involved three sample sets. The first sample set consisted of paired tumor, tumor-adjacent, and tumor-distant tissues from 25 breast cancer patients, and control breast tissue samples from 10 cancer-free women. The second sample set comprised 16 commercial breast cancer and two non-cancerous breast cell lines. For the third sample set, cell lines MCF-7, BT-474, SK-BR3, and MDA-MB-231 were treated with doxorubicin (Dox) or paclitaxel (Tax) for 72 h. In addition to the analysis of epigenetic alterations, alternative splicing, and gene expression, sample set 3 was subjected to mass spectrometry-based shotgun proteomics in order to reveal global changes in protein expression upon Dox or Tax treatment.

## 2. Materials and Methods

### 2.1. Samples

#### 2.1.1. Breast Tissue Samples

Breast tissue samples were collected from 25 breast cancer patients at diagnosis. The study was approved by the Ethics Commission of the Medical University of Vienna (applications EK 2003/260 and EK 2003/366). All women gave permission for sampling by signing a written informed consent form. Using ultrasound-guided needle biopsies, three samples were taken from each patient: tumor tissue from the tumor center, tumor-adjacent tissue from histologically normal tissue (distance from the tumor center about 1 cm), and tumor-distant tissue from histologically normal tissue (distance from the tumor center about 3 cm). However, tumor-adjacent tissue from patient 10 was not available.

Breast cancer patients were between 32 and 86 years old (mean 58.4 years, median 59.0 years). None of the women had a family history of breast cancer. In addition, none of the patients had received radiotherapy, chemotherapy, or hormonal treatment.

Table 1 gives information on the characteristics of the breast cancer patients, including age, menopausal status, histological type, histological grading, B classification, proliferative activity (MIB-1), status of estrogen receptor (ER), progesterone receptor (PR), and Her2, as well as the molecular subtype.

Breast tissue samples from ten cancer-free women who underwent breast reduction mammoplasty served as controls. The age of these women ranged from 20 to 59 years (mean 36.9 years, median 34.5 years). Tissue samples were stored in phosphate-buffered saline (PBS) at −80 °C until DNA extraction.

#### 2.1.2. Cell Lines

The study used breast cancer cell lines with the molecular subtype luminal A: CAMA-1, KPL-1, MCF-7, T47D, and ZR-75-1; luminal B: BT-474; Her2-positive: AU-565 and SK-BR3 (derived from the same patient); triple-negative: BT-549, Cal-51, HCC 1143, HCC 1937, HS 578T, MDA-MB-231, MDA-MB-453, MDA-MB-468; and MCF 10A and MCF 10F, two non-cancerous breast cell lines from fibrocystic disease.

Cells were obtained either from the ATCC (American Type Culture Collection; Manassas, VA, USA) or the DSMZ (German Collection of Microorganisms and Cell Cultures GmbH; Braunschweig, Germany). ATCC and DMSZ authenticate all cell lines by short tandem repeat (STR) profiling and other methods before distribution.

Cells were cultured at 37 °C in a humidified 5% CO_2_ incubator as described previously [41]. In brief, most cell lines were grown in Roswell Park Memorial Institute (RPMI) 1640 media (Invitrogen, Vienna, Austria) supplemented with 5% fetal calf serum (FCS) (Invitrogen, Vienna, Austria), except AU-565 and MDA-MB-453 (RPMI 1640 with 10% FCS) as well as BT-474, HS 578T, MCF 10A, and MCF 10F, which were each cultured in different media, as described in [41].

Cells were harvested at ~70% confluency and cell pellets were stored at −80 °C until DNA extraction. CAMA-1, KPL-1, MCF-7, T47D, BT-474, SK-BR3, BT-549, Cal-51, HCC 1143, HCC 1937, MDA-MB-231, MDA-MB-453, and MCF 10A breast cell lines were additionally harvested with Tri Reagent (Sigma-Aldrich, Schnelldorf, Germany) and immediately lysed for direct RNA extraction.

#### 2.1.3. Treatment of MCF-7, BT-474, SK-BR3, and MDA-MB-231 Cells with Dox or Tax

To determine sensitivity to Dox (dissolved in 0.9% NaCl solution, Sigma-Aldrich) or Tax (Sigma-Aldrich), MCF-7, BT-474, SK-BR3, and MDA-MB-231 cells were seeded in 96-well plates at a density of 3500–7000 cells for 72 h in the respective media containing 10% FCS. Cells were treated with increasing drug concentrations on the following day. Cell viability was determined by the 3-(4,5-dimethylthiazol-2-yl)-2,5-diphenyltetrazolium bromide (MTT)-based vitality assay (EZ4U; Biomedica, Vienna, Austria) following the manufacturer´s instructions. IC_50_ values (drug concentration that induces reduction in cell number to 50%) were determined using full dose–response curves by using GraphPad Prism software (version 8.0.1) and are based on three biological replicates, with each running in three technical replicates.

For analyzing the effects from Dox or Tax treatment, 1–2.5 million cells were seeded in T75 cm^2^ cell culture flasks and treated with the respective IC_50_ concentrations of Dox (MCF-7: 704.3 nM, BT-474: 514.9 nM, SK-BR3: 643.3 nM, MDA-MB-231: 645.8 nM) and Tax (MCF-7: 1.9 nM, BT-474: 1.9 nM, SK-BR3: 4.9 nM, MDA-MB-231: 56.5 nM) for 72 h on the following day. An untreated control from each cell line was included. After treatment, cells were directly processed for protein extraction (Section 2.5). For DNA and RNA extraction, cells were trypsinized, washed twice with PBS, and pelleted by centrifugation at 300× *g* for 5–10 min. RNAprotect cell reagent (Qiagen, Hilden, Germany) was added to the cell pellet according to the manufacturer’s instructions. Pellets were stored at −20 °C until further processing. The treatment experiment was performed in triplicate on different days.

### 2.2. DNA and RNA Extraction

From breast tissues, aliquots ≤ 25 mg were taken, cut into small pieces and lysed for 4 h. DNA was isolated using a QIAamp DNA Mini Kit (Qiagen), following the manufacturer´s protocol for tissue.

From commercial breast cell lines, genomic DNA was isolated using a High Pure PCR Template Preparation Kit (Roche, Vienna, Austria) according to the manufacturer´s instructions. For BT-549, MCF 10A, MDA-MB-468, and a smart DNA prep (m) kit (Analytik Jena, Jena, Germany) was used. Total RNA from cell lines was purified using the RNeasy Mini Kit (Qiagen), and subjected to quality control as described previously [41].

For treated cell lines and untreated controls, DNA and RNA were isolated simultaneously using an AllPrep DNA/RNA Mini Kit (Qiagen) following the manufacturer´s protocol for animal cells using 40 μL of 1 M dithiothreitol (DTT; Qiagen) per 1 mL Buffer RLT Plus for cell lysis. Additional on-column DNase digestion was performed according to the appendix E protocol using a RNase-Free DNase Set (Qiagen).

Isolated DNA was quantified using either the Qubit dsDNA HS Assay Kit or the Qubit dsDNA BR Assay Kit with the Qubit 4 instrument. Isolated RNA was quantified using the Qubit RNA BR Assay Kit (all Thermo Fisher Scientific, Vienna, Austria). RNA integrity and quality were checked using the Qubit RNA IQ Assay Kit (Thermo Fisher Scientific, Austria). For all RNA extracts, an RNA IQ score ≥ 7.8 was obtained, indicating that the extracts contained ≥78% large and/or structured RNA and ≤22% small RNA. DNA and RNA extracts were stored at −20 °C.

### 2.3. DNA Methylation Analysis

For sodium bisulfite conversion of unmethylated cytosines, an EpiTect Fast Bisulfite Conversion Kit (Qiagen, Germany) was used according to the manufacturer’s protocol. Converted DNA was quantified with a Qubit ssDNA Assay Kit (Thermo Fisher Scientific, Austria).

PSQ assays were developed in-house. The nucleotide sequence of *ABCA7* (GenBank: NG_046909.1) and exon/intron annotation were taken from the National Center for Biotechnology Information (NCBI) database [42], and the location of the promoter was taken from Iwamoto et al. [9]. Primers were designed with PyroMark Assay Design Software 2.0.1.15 (Qiagen). SNPs discovered by the 1000 Genomes Project [43] and located in the target region were taken into consideration. However, our focus was to investigate CpG methylation, and only three of these SNPs in the promoter could be analyzed automatically without impairing CpG analysis by the PyroMark Q24 Advanced software. In order to enable DNA methylation analysis of DNA strands carrying minor alleles of other SNPs located in the sequence to analyze, we adapted the dispensation order by including the respective nucleotides if possible. Primers were purchased from Sigma-Aldrich (Steinheim, Germany). For each primer set, concentration and annealing temperature (T_a_) were optimized by using bisulfite-converted unmethylated and methylated control DNA. Both commercially available and unmethylated and methylated human control DNA prepared in-house was used. Commercially available human control DNA included EpiTect Control DNA (Qiagen) and CpGenome Universal Methylated DNA (Merck, Germany). In-house preparation was done as described previously [44]. In brief, for generating unmethylated control DNA, at least 24 ng DNA isolated from normal breast tissue was amplified by whole-genome amplification using an REPLI-g Mini Kit (Qiagen) and purified by sodium acetate precipitation. Methylated control DNA was obtained by subsequent CpG Methyltransferase M.SssI (NEB, Frankfurt am Main, Germany) treatment followed by sodium acetate precipitation.

In total, we targeted 45 CpGs, located in six different regions of the *ABCA7* gene, by applying primer sets prom, A, B, C, D, and E (Figure 1). Table 2 summarizes the sequences for each primer set, the length of the resulting amplicons, the number of CpGs covered by sequencing, and the experimental PCR conditions. For primer sets A, B, C, D and E, PCR was carried out with a PyroMark PCR Kit (Qiagen). Each reaction was performed in a total volume of 25 µL, consisting of 12.5 µL PyroMark PCR Master Mix (2×), 2.5 µL CoralLoad Concentrate (10×), RNase-free water, forward and reverse primer (final concentrations are given in Table 2), and 10 ng of bisulfite-converted DNA, while 0.5 mM additional MgCl_2_ was used for primer set D. For amplification, either iCycler (Bio-Rad, Hercules, CA, USA) or QuantStudio 5 (Thermo Fisher Scientific) was used. The temperature program was as follows: initial activation at 95 °C for 15 min; 50 cycles: 30 s at 94 °C, 30 s at the respective annealing temperature (Table 2), 30 s at 72 °C; and final extension at 72 °C for 10 min. For primer set prom, a Type-it HRM PCR Kit (Qiagen) was used. Each reaction was performed in a total volume of 20 µL, consisting of 10 µL Type-it HRM PCR (2×), RNase-free water, forward and reverse primer each in a final concentration of 200 nM, and 10 ng of bisulfite-converted DNA. For amplification, a Rotor-Gene Q (Qiagen) was used with the temperature program as follows: initial activation at 95 °C for 5 min; 50 cycles: 10 s at 94 °C, 20 s at 58.5 °C, 20 s at 72 °C; and final extension at 72 °C for 10 min. A control lacking template DNA was included in each run, serving as a control for contamination.

Identity and purity of the PCR products were assessed by gel electrophoresis (3% agarose gel in Tris–borate–EDTA (TBE) buffer. The gel was post-stained with 3× GelRed (Biotium, Fremont, CA, USA) and bands were visualized with an UVT-20 M transilluminator (Herolab, Germany).

PSQ analyses were performed using a PyroMark Q24 Vacuum Workstation and PyroMark Q24 Advanced instrument with PyroMark Q24 Advanced CpG Reagents (all Qiagen) according to the manufacturer´s instructions. In brief, for primer sets A, B, C, D, and E, 1 μL Streptavidin Sepharose High Performance (GE Healthcare, Germany), 40.0 μL PyroMark Binding Buffer, 24.0 µL high-purity water (Milli-Q 18.2 MΩ), and 15.0 μL of biotinylated PCR product were mixed by agitating for 10 min at 1400 rpm. For primer set prom, 1.3 μL Streptavidin Sepharose High Performance (GE Healthcare, Germany), 50.7 μL PyroMark Binding Buffer and 30.4 µL high-purity water (Milli-Q 18.2 MΩ) were mixed with 19.0 μL of biotinylated PCR product. The double-stranded DNA was denatured, washed and finally transferred into a PyroMark Q24 Plate (Qiagen) containing 20.0 µL of 0.375 μM sequencing primer. After an incubation step of 5 min at 80 °C on a preheated PyroMark Q24 Plate Holder (Qiagen), the plate was placed into the instrument and sequencing was initiated. PSQ data was evaluated with the PyroMark Q24 Advanced software 3.0.0 (Qiagen). A representative pyrogram for each primer set is shown in (Appendix A).

### 2.4. Gene Expression and Analysis of Intron-Containing Transcripts

Reverse transcription of RNA was performed with a QuantiTect Reverse Transcription Kit (Qiagen) following the manufacturer’s instructions.

All qPCR assays were developed in-house using the nucleotide sequence of *ABCA7* (GenBank: NG_046909.1) and exon/intron annotation from the NCBI database [42]. *PUM1* (GenBank: NM_001020658.2) was chosen as house-keeping gene as suggested previously [45,46]. Primers were designed with the web interface Primer3Plus [47] and purchased from Sigma-Aldrich (Steinheim, Germany).

*ABCA7* mRNA transcript levels were determined by qPCR with a QuantiTect SYBR Green PCR Kit (Qiagen) and exon-specific primer sets (Table 3), each containing one primer spanning the respective exon–exon boundary. Each reaction was performed in a total volume of 20 µL, consisting of 10 µL QuantiTect SYBR Green PCR Master Mix, RNase-free water, forward and reverse primer (final concentration of 300 nM each), and 10 ng cDNA using the QuantStudio 5 instrument (Thermo Fisher Scientific) and fast ramp speed. The temperature program included an initial activation step of 15 min at 95 °C, followed by a 40× repeated 3-step cycling of 10 s denaturation at 94 °C, annealing for 20 s at 60 °C, and elongation for 20 s at 72 °C. For primer set *ABCA7* exon 3–exon 4, 35 cycles were used. Subsequent melt curve acquisition was performed from 60 °C to 95 °C in 0.1 °C/s steps.

For analysis of intron-containing transcripts, qPCR was carried out with the QuantiTect SYBR Green PCR Kit (Qiagen) and intron-specific primer sets (Table 4), each having one primer located in the respective exon and the second primer in the following intron. qPCR was performed as described for gene expression analysis using the following temperature program: initial activation at 95 °C for 15 min; 45 cycles: 10 s at 94 °C, 20 s at 60 °C, 20 s at 72 °C. Specificity of PCR was checked by subsequent melt curve analysis.

Intron retention was verified by performing qPCR with the QuantiTect SYBR Green PCR Kit (Qiagen) and intron-flanking primer sets (Table 5), with the forward primer targeting the exon upstream and the reverse primer targeting the exon downstream of the respective intron. qPCR was carried out as described for gene expression analysis using the following temperature program: initial activation at 95 °C for 15 min; 45 cycles: 10 s at 94 °C, 20 s at 60 °C, 30 s at 72 °C. Melt curve analysis was performed to check for the presence of multiple PCR products.

Controls lacking template DNA or reverse transcriptase were included in each qPCR run to rule out general and gDNA contamination, respectively.

For each qPCR assay, the identity and purity of the PCR products were checked by gel electrophoresis (3% agarose gel in TBE buffer and post-staining using 3× GelRed (Biotium, USA) and a UVT-20 M transilluminator (Herolab, Germany)).

### 2.5. Mass Spectrometry-Based Shotgun Proteomics

*Fractionation.* The treated MCF-7, BT-474, SK-BR3, and MDA-MB-231 cells were processed according to a nucleocytoplasmic fractionation protocol as previously described [48]. First, the cells were washed with PBS. One milliliter isotonic lysis buffer (10 mM sodium 2-[4-(2-hydroxyethyl)piperazin-1-yl]ethane-1-sulfonate (HEPES), 10 mM NaCl, 3.5 mM MgCl_2_, 1 mM ethylene glycol-bis(β-aminoethyl ether)-N,N,N′,N′-tetraacetic acid (EGTA), 0.25 M Sucrose, 0.5% Triton X-100), which contained protease inhibitors (1% phenylmethylsulfonyl fluoride (PMSF) and 1% protease and phosphatase inhibitor cocktail from Roche), was added. The cells were scraped and transferred into Falcon tubes (15 mL, Corning). The membrane was ruptured using shear stress by a syringe. After centrifugation (1200× *g*, 5 min, 4 °C), the supernatant was transferred into ice-cold ethanol (1:5) and stored at −20 °C overnight (=CYT fraction). The pellet containing the nuclei was incubated in 900 µL hypertonic solution (10 mM Tris-HCl, 1 mM EDTA and 0.5 M NaCl) and 100 µL NP-40 buffer (10 mm Tris-HCl, 1 mm EDTA and 0.5% Triton X-100) containing protease inhibitors (1% PMSF from Sigma and 1% protease and phosphatase inhibitor cocktail from Roche) was added. After centrifugation (1200× *g*, 5 min), the soluble nuclear proteins were also transferred into ice-cold ethanol (1:5) and stored at −20 °C overnight (=NE fraction). The precipitated proteins were pelleted, dried under vacuum and dissolved in sample buffer (100 mM DTT, 7.5 M urea, 1.5 M thiourea, 4% CHAPS, 0.05% sodium dodecyl sulphate) and the protein concentration was determined using a BCA assay.

*Digestion protocol.* For enzymatic protein digestion, 20 µg of protein was used and the ProtiFi S-trap technology applied [49]. Briefly, solubilized protein was reduced and carbamidomethylated by adding 64 mM DTT and 48 mM iodoacetamide (IAA), respectively. Prior to sample loading onto the S-trap mini cartridges, trapping buffer (90% *v*/*v* methanol, 0.1 M triethylammonium bicarbonate) was added. Samples were thoroughly washed and digested with Trypsin/Lys-C Mix at 37 °C for 2 h (1:40 ratio). Finally, peptides were eluted, dried and stored at −20 °C.

*LC-MS/MS analysis.* The dried peptide samples were reconstituted in formic acid (5 µL, 30%), which contained four synthetic peptides (at 10 fmol) as external standards. Then, loading buffer (40 µL, 97.9% H_2_O, 2% acetonitrile, 0.05% trifluoroacetic acid) was added, vortexed and put into the autosampler. The samples were analyzed using a nano-LC-MS/MS system consisting of a Dionex Ultimate 3000 nano-LC (Thermo Fisher Scientific, Bremen, Germany) hyphenated to a timsTOF Pro mass spectrometer (Bruker Daltonics, Bremen, Germany). The samples were concentrated on a Pepmap100 pre-column (2 cm × 100 μm C18, Thermo Fisher Scientific) and chromatographic separation was carried out on an Aurora emitter column (25 cm × 75 μm C18, Ionoptics). The sample volume was 1 µL and the flow rate on the emitter column was 300 nL·min^−1^. Eluent A consisted of 99.9% H_2_O, 0.1% formic acid and eluent B consisted of 80% acetonitrile, 20% H_2_O, 0.1% formic acid. The captive ion spray source was run at 1.6 kV. We used the PASEF mode (10 PASEF cycles), a cycle time of 1.16 s and a target intensity threshold of 2500.

*Gradient of response profiling.* The analysis consisted of a 90 min gradient from 8% to 40% eluent B. The total run time was 135 min per sample.

*Data evaluation.* MaxQuant (Version 1.6.17.0), including the in-built Andromeda search engine, was used for label-free quantification of proteomic raw data. Non-redundant Swissprot entries were used with at least two peptides for identification, of which one needed to be unique. The first and main search peptide tolerances were 50 and 25 ppm, respectively. The false-discovery rate (FDR) was fixed to 0.01 on the peptide and protein level and maximum two missed cleavages. Furthermore, search criteria included carbamidomethylation of cysteine as fixed modification and methionine oxidation, as well as N-terminal acetylation as dynamic modifications. Proteins were considered if they showed at least one unique peptide. Statistical evaluation was performed with Perseus software (1.6.10.43) using LFQ intensities of the MaxQuant result files. The option “match between runs” was selected. Permutation-based FDR set to 0.05 and S0 = 0.1 was used for *t*-tests and gave multiparameter-corrected significant protein expression changes.

### 2.6. Bioinformatics

Genome-wide methylation data (Illumina Infinium Human Methylation 450K BeadChip) including clinical information was downloaded from TCGA using function getTCGA from R-package TCGA2STAT 1.2 (http://www.liuzlab.org/TCGA2STAT/ accessed on 15 January 2022). Breast cancer PAM50 molecular subtype information [50] was retrieved with Bioconductor R-package TCGAbiolinks 2.14.0 [51]. Beta values for cg06730721, cg26263477, cg05504606, cg26264438, cg07726048, cg21590311, cg10749413, cg05989429, cg23027715, cg02817925, cg06169110, cg02253236, cg02959678, cg02807077, cg10406526, cg00874873, cg18529892, cg24145486, cg02986791, cg05372495, cg12082025, cg26576206, cg13761375, cg11714200, cg19548313, and cg18073514 and molecular subtype information were available for 683 breast cancer patients.

Splicing factor binding sites were predicted using RBPmap [52]. Binding sites were predicted (high stringency level) for introns 3, 5, and 6 as well as for 50 bp-long sequences in the adjacent exons.

### 2.7. Statistical Analysis

Statistical analyses were performed with either IBM SPSS Statistics 26.0 or R-package rstatix 0.7.0 [53]. DNA methylation levels <5% (lower limit of quantification, LLOQ) and >95% (upper limit of quantification, ULOQ) were substituted with default values, namely, 2.5% and 97.5%, respectively, as proposed previously [54]. For RNA levels, relative quantification was performed using the ΔΔCT (cycle threshold) method. In brief, ΔCT was calculated as difference between CT values of the target gene *ABCA7* to the reference gene *PUM1* for each sample and *ABCA7* primer set individually. For comparison between different assays, difference was then calculated for each sample to the ΔCT level of *ABCA7* exon 5–exon 6 of reference sample MDA-MB-231 of the respective sample set (2 or 3). Negative ΔΔCT values were chosen for graphical representation because higher −ΔΔCT values indicate higher transcript levels. One-way ANOVA (analysis of variance), followed by post hoc Tukey´s HSD (honest significant difference) test was applied to test for significant differences in the DNA methylation status and RNA levels between tumors of different molecular subtypes; between tumor, tumor-adjacent, and tumor-distant tissues and breast tissues from healthy controls; between different genotypes; and between treatment and control, where applicable.

Differences in beta values between breast cancer molecular subtypes were calculated with the R-package multcomp 1.4-12 [55] using two-way ANOVA followed by Tukey´s all-pair comparisons.

Groups consisting of only one member were excluded from testing. Pearson’s correlation coefficient was used to assess the relationship between RNA levels and DNA methylation or between RNA levels from different assays. Pearson’s correlation coefficient was also used to assess the relationship between DNA methylation levels and age and cell proliferation. In all cases, a *p* value <0.05 (two-sided) was considered statistically significant.

## 3. Results

### 3.1. DNA Methylation Data Retrieved from TCGA

DNA methylation data, determined with the Infinium Human Methylation 450K BeadChip, was retrieved from TCGA for 683 breast cancer tissues samples. Six CpGs in the *ABCA7* promoter region and two CpGs in intron 1 (Figure 1) showed beta values <0.2 (Figure 2a and Appendix A). CpGs located downstream of exon 12 and upstream of exon 45 were rather highly methylated in all molecular subtypes (mean beta values >0.75) with the exception of CpGs in exon 14 for basal-like and normal-like samples, and CpGs in intron 22, which were heterogeneously methylated (Appendix A). In contrast, two CpGs in exon 46 and intron 46, respectively, showed low (mean beta values <0.2) methylation levels, whereas three CpGs downstream of exon 47 were heterogeneously methylated.

### 3.2. Breast Tissue Samples

Paired tumor, tumor-adjacent, and tumor-distant tissues from 25 breast cancer patients and control breast tissue samples from 10 cancer-free women were subjected to DNA methylation and SNP analysis in order to identify breast cancer-specific aberrations. Analysis of this sample set should yield information on the occurrence of epigenetic aberrations in tumor-adjacent tissues that appear histologically normal.

#### 3.2.1. DNA Methylation Status of CpGs in ABCA7 Promoter–Exon 1 and Intron 1

By applying our primer sets for the promoter–exon 1 region and the intron 1 regions A and B (Figure 1) to six tumor tissue (luminal A: T7, T8; luminal B: T10, T17; Her2-positive: T18; triple-negative: T14) and four control tissue samples (C1–C4), all CpGs showed methylation levels <15% (Figure 2b). CpG 2 in region A follows a homopolymeric stretch of six thymines, causing overestimation of the methylation status. Thus, methylation data for CpG 2 in region A was excluded from data evaluation. Since our results did not suggest breast cancer-specific DNA methylation alterations in promoter–exon 1 and intron 1, we refrained from determining the methylation status of these regions for a higher number of breast tissue samples.

#### 3.2.2. Genotypes of SNPs rs531508435, rs549725064, and rs3764642

The sequence to analyze of the PSQ assay targeting the promoter–exon 1 region allowed automatic analysis of three SNPs. We performed SNP genotyping for several tumor (luminal A: T7, T8; luminal B: T10, T17; Her2-positive: T18; triple-negative: T14) and control samples (C1–C5, C7). For the SNPs rs531508435 (NC_000019.10: g.1040065G>A MAF: <0.001(A); G>T MAF: 0.000(T)) and rs549725064 (NC_000019.10: g.1040081G>A MAF: <0.001(A)), we only detected the major alleles (Appendix A). However, for rs3764642 (NC_000019.10: g.1040047G>A, MAF: 0.448(A); G>C, MAF: 0.000(C)), we detected the heterozygous G/A variant in seven out of 12 tissue samples. The homozygous A variant was found in four out of 12 tissue samples, the homozygous G variant only in one out of 12 tissue samples. The A allele was detected in tumor and control tissues, indicating that its occurrence was not breast cancer-specific. We therefore refrained from subjecting a higher number of breast tissue samples to SNP genotyping.

#### 3.2.3. DNA Methylation Status of CpGs at ABCA7 Exon 5–Intron 5 Boundary

In tumor tissues from breast cancer patients (Figure 3a–d) and normal tissues from cancer-free women (Figure 3e), CpGs at the *ABCA7* exon 5–intron 5 boundary (region C) were found to be more highly methylated than CpGs in region A and B of intron 1. In all tumor tissues, CpG 5, the intronic CpG closest to the exon 5–intron 5 boundary, showed a higher DNA methylation status than downstream CpGs 6 and 7.

In luminal A tumors, each of the seven CpGs was significantly more highly methylated than in triple-negative tumors and normal breast tissues from cancer-free women serving as controls (Figure 4). In addition, CpG 2 (Figure 4b) and CpG 5 (Figure 4e) were significantly more highly methylated in tumors of the luminal B subtype than in triple-negative ones. For CpG 1–5 (Figure 4), a significant difference was also found between luminal B tumors and normal breast tissues from cancer-free women.

In order to elucidate if *ABCA7* methylation status at the exon 5–intron 5 boundary is a marker for epigenetic field cancerization, we also determined the methylation status for tumor-adjacent and tumor-distant tissues (Figure 4) from these breast cancer patients. For patients with luminal A breast cancer, not only the tumors but also tumor-adjacent and tumor-distant tissues showed a significantly higher DNA methylation status than normal breast tissues from controls. Each of the seven CpGs was significantly more highly methylated in tumor tissues than in paired tumor-adjacent tissues and tumor-distant tissues. No significant difference in methylation status was, however, observed between tumor-adjacent and tumor-distant tissues (Figure 4).

For CpGs 1–5, a significant difference in methylation status was found between tumors of the luminal B subtype and control breast tissues. No significant difference was, however, observed between the respective tumor-adjacent and tumor-distant tissues and control tissues.

In tumors of the triple-negative subtype, the methylation status of CpGs 1–7 was generally lower than in the respective tumor-adjacent and tumor-distant tissues. For CpG 2 and CpG 5, the difference between triple-negative tumor tissues and the respective tumor-distant tissues was significant.

#### 3.2.4. Association of the DNA Methylation Status at the Exon 5–Intron 5 Boundary in Breast Tissues with Cell Proliferation and Age

We investigated if the *ABCA7* DNA methylation status in tumor tissues is associated with expression of Ki-67, a cellular marker for proliferation. For none of the seven CpGs at the exon 5–intron 5 boundary did we find DNA methylation status in the tumor to be correlated with Ki-67 expression, independently of whether tumor samples of different molecular subtypes were pooled or tested separately. Neither for breast cancer patients nor for cancer-free women did we find the DNA methylation status at the exon 5–intron 5 boundary to be associated with the age of the probands.

### 3.3. Commercial Breast Cell Lines

Since RNA was not available from breast tissue samples, sample set 2, comprising 16 commercial breast cancer and two non-cancerous breast cell lines, was analyzed to detect any potential association between DNA methylation, SNP genotypes, gene expression, and alternative splicing of *ABCA7*.

#### 3.3.1. DNA Methylation Status of CpGs in Promoter–Exon 1 and Intron 1

By determining the DNA methylation status of the CpGs in the *ABCA7* promoter–exon 1 region and the intron 1 regions A and B (Figure 1) for 14 breast cell lines (luminal A: CAMA-1, KPL-1, MCF-7, T47D, and ZR-75-1; luminal B: BT-474; Her2-positive: AU-565 and SK-BR3; triple-negative: HCC 1143, HCC 1937, HS 578T, MDA-MB-231, MDA-MB-453; and non-cancerous: MCF 10F), eight CpGs in promoter–exon 1, four CpGs in exon 1, and eight CpGs in intron 1 showed methylation levels <20% (Figure 5a).

Higher methylation levels were only observed for HCC 1937 in region A (CpG 5, 23.6%) and region B (CpG, 1 22.0%; CpG 2, 22.8%; CpG 3, 19.2%; and CpG 4, 32.5%).

#### 3.3.2. Genotypes of SNPs rs531508435, rs549725064, and rs3764642

Only the major alleles of the SNPs rs531508435 and rs549725064 were detected in commercial breast cell lines. For rs3764642, we detected the heterozygous G/A variant in seven out of 18 cell lines (Figure 5b). The homozygous A variant was found in seven out of 18 cell lines, and the homozygous G variant only in four out of 18 cell lines. In line with our results for breast tissues, occurrence of the A allele was not breast cancer-specific.

#### 3.3.3. DNA Methylation Status of CpGs at Exon 5–Intron 5 Boundary

Among the three *ABCA7* regions investigated, the biggest differences in the DNA methylation status between the breast cell lines were detected at the exon 5–intron 5 boundary (Figure 6). In all luminal A (Figure 6a), in six out of eight triple-negative (Figure 6d) and in both non-cancerous breast cell lines (Figure 6e), the methylation status of the seven CpGs was >74%. The two Her2-positive cell lines showed an intermediate methylation level (between 13% and 66%, Figure 6c). In the luminal B cell line, the seven CpGs were largely unmethylated (all CpGs <LLOQ, except CpG 5, 9%; Figure 6b). In all cell lines, CpG 6 and CpG 7 were less methylated than CpG 5. In the Her2-positive subtype, the methylation status of CpGs 1, 2, and 5 was significantly lower than in luminal A and triple-negative and non-cancerous breast cell lines serving as controls (Figure 6f).

#### 3.3.4. DNA Methylation Status of CpGs in ABCA7 Exon 12

Several cell lines (luminal A: MCF-7, T47D, and ZR-75-1; luminal B: BT-474; Her2-positive: SK-BR3; triple-negative: MDA-MB-231, MDA-MB-453) were screened for their DNA methylation status in *ABCA7* exon 12 by applying assay E. The eight CpGs targeted were highly methylated (>75%), except CpG 7 in Her2-positive cell lines (Appendix A). In 674 out of 683 TCGA breast cancer tissues samples, cg23027715 (corresponding to CpG 7), also showed a DNA methylation status >75% (Appendix A).

#### 3.3.5. ABAC7 mRNA Levels

RNA was available from 13 breast cell lines, including 12 cancer and one non-cancerous cell line. Numerous transcript variants of *ABCA7* are known [42,56], including those annotated by GENCODE 43. Only transcripts starting from exon 1 or exon 2 (Figure 1) should result in the functional ABCA7 transporter, consisting of two transmembrane domains and two ATP-binding domains (Figure 7, ABCA7-201).

We determined expression levels for seven exon–exon boundaries (exon 1–exon 7, exon 18–19; Table 3) by using in-house developed exon–exon boundary-specific qPCR assays.

In one and the same cell line, abundance of the seven exon–exon boundaries differed substantially. In general, exon 1–2 boundary levels were about a factor of 2 lower than levels of the exon 2–3 boundary (Figure 8). Exon 1 was also expressed slightly less than exon 3–4, 4–5, and 5–6 boundaries in the cell lines BT-474, MCF 10A, and CAL-51.

In SK-BR3, the exon 18–19 boundary was the most abundant exon–exon boundary investigated. In MCF 10A, the exon 18–19 boundary level was similar to those of the other exon–exon boundaries, except exon boundaries 2–3 and 3–4, which were more highly expressed. In BT-474, the exon 18–19 boundary was found to be more abundant than the other exon–exon boundaries, except the exon 2–3 and exon 5–6 boundaries. In luminal A and triple-negative cell lines, the difference in the expression of exon 18–19 and the other exon–exon boundaries was quite diverse. In CAMA-1, the exon 18–19 boundary was less, and in T74D more abundant than the other exon–exon boundaries investigated. In KPL-1 and MCF-7, the exon 18–19 boundary was more highly expressed than the other exon–exon boundaries, except the exon 2–3 boundary. In BT-549, CAL-51, and MDA-MB-231, the exon 18–19 boundary level was generally less or equally abundant compared to the other exon–exon boundaries, whereas for HCC 1143 and HCC 1937, the exon 18–19 boundary was found to be more abundant than the exon 1–2, exon 4–5, exon 5–6, and exon 6–7 boundaries. In all cell lines, the exon 6–7 boundary was less abundant than the exon 2–3 and 3–4 boundaries. We did not find significant differences in *ABCA7* exon levels between different molecular subtypes of breast cancer (*p* > 0.05). In addition, *ABCA7* exon levels did not correlate with DNA methylation levels of CpGs in promoter–exon 1, intron 1, or at the exon 5–intron 5 boundary.

#### 3.3.6. Intron-Containing Transcripts

Next, we searched for known intron-containing transcripts of ABCA7. Transcripts containing introns may result from either intron retention (e.g., intron 6 in ABCA7-202) or alternative intron termination (e.g., intron 6 in ABCA7-208, Figure 7). By applying qPCR assays involving intron-specific primer sets with one primer located in the exon and the second one in the consecutive intron (Table 4), transcripts containing introns 3–7 were observed for each of the 13 cell lines (Figure 8). We verified intron retention by applying qPCR assays developed in-house using intron-flanking primer sets (Table 5). By subjecting the PCR products obtained with intron-flanking primer sets to agarose gel electrophoresis, we detected PCR products containing intron 3, 4, 5, 6, or 7 in between its flanking exons in addition to PCR products lacking the respective intron (Figure 9a–e). The presence of PCR products containing the respective intron was confirmed by melt curve analysis (Figure 9f, shown for intron 6). However, alternative intron termination or alternative intron start was only detected for intron 7, but not for intron 3, 4, 5, or 6 (Appendix A). Presumably, the concentration of PCR products containing intron 3, 4, 5, or 6 was below limit of detection (LOD) of agarose gel electrophoresis.

Intron 6 was found to be more abundant than the other introns investigated. In addition, intron 6 levels were quite as high as those of the exon–exon boundaries. The difference between the expression levels of the exon 6–7 boundary and the exon 5–6 boundary correlated with the expression level of intron 6 (*r* = 0.81, *p* < 0.001), indicating that the lower abundance of the exon 6–7 boundary was linked to higher retention or alternative termination of intron 6.

The extent of intron retention/alternative intron termination was not associated with the DNA methylation status at the exon 5–intron 5 boundary. In order to investigate if there is an association between intron 6 retention/alternative termination and the methylation status at the exon 6–intron 6 boundary, we screened seven cell lines (MCF-7, T47D, ZR-75-1, BT-474, SK-BR3, MDA-MB-231, and MDA-MB-468) for their DNA methylation status at the exon 6–intron 6 boundary by targeting five CpGs in exon 6 and four in intron 6 (Appendix A). Our results did not hint at any association between intron 6 retention/alternative termination and the DNA methylation status at the exon 6–intron 6 boundary. In addition, no direct association between intron 6 retention/alternative termination and the molecular subtype was found (Figure 8f).

### 3.4. Breast Cancer Cell Lines Treated with Dox or Tax

In order to investigate if chemotherapeutic treatment is associated with alterations in ABCA7, MCF-7, BT-474, SK-BR3, and MDA-MB-231, representing luminal A, luminal B, Her2-positive, and triple-negative breast cancer, respectively, were treated for 72 h at IC50 with Dox (MCF-7: 704.3 nM, BT-474: 514.9 nM, SK-BR3: 643.3 nM, MDA-MB-231: 645.8 nM) or Tax (MCF-7: 1.9 nM, BT-474: 1.9 nM, SK-BR3: 4.9 nM, MDA-MB-231: 56.5 nM). The IC_50_ values indicated that the sensitivity of the four cells for Dox was rather similar, while that for Tax was quite different.

Treated cells and untreated cells as controls were investigated for specific alterations in DNA methylation, expression, and intron retention/alternative intron termination of *ABCA7*. In order to reveal global changes in protein expression upon Dox or Tax treatment, we also performed mass spectrometry-based shotgun proteomics by analyzing NE and CYT fractions separately.

#### 3.4.1. DNA Methylation Status of CpGs at the Exon 5–Intron 5 Boundary

Cancer cells surviving Dox or Tax treatment did not differ in the DNA methylation status of the CpGs at the exon 5–intron 5 boundary from untreated cells (Appendix A).

#### 3.4.2. ABCA7 mRNA Levels

Dox treatment resulted in significant alterations in *ABCA7* mRNA levels (Figure 10), depending on the molecular subtype. In MCF-7 and BT-474 cells surviving Dox treatment, *ABCA7* was significantly up- (Figure 10a) and down-regulated (Figure 10b), respectively. In SK-BR3 cells surviving incubation with Dox, decreased expression of the exon 18–19 boundary was observed (Figure 10c), whereas in MDA-MB-231 cells, Dox did not have an impact on *ABCA7* expression (Figure 10d). Tax treatment affected only *ABCA7* expression in SK-BR3 cells, resulting in a significantly down-regulated exon 18–19 boundary compared to untreated cells (Figure 10c).

#### 3.4.3. Intron-Containing Transcripts

When we investigated the impact of Dox or Tax on alternative splicing, we detected molecular subtype- and intron-specific events (Figure 10). Dox resulted in increased intron 3 and intron 6 retention/alternative termination in MCF-7 cells (Figure 10a). For SK-BR3, we observed significantly increased intron 3 and intron 5 retention/alternative termination (Figure 10c). MDA-MB-231 cells surviving Dox incubation showed increased intron 6 retention/alternative termination (Figure 10d). Tax treatment of MCF-7 and SK-BR3 cells resulted in significant decrease (Figure 10a) and significant increase (Figure 10c) in intron 5 levels, respectively. Incubating MDA-MB-231 cells with Tax did not have an effect on retention/alternative termination of the introns investigated (Figure 10c). For BT-474, neither Dox nor Tax altered the extent of intron retention/alternative intron termination compared to untreated cells (Figure 10b).

Neither *ABCA7* transcript levels nor the formation of *ABCA7* splicing variants was found to be associated with the DNA methylation status at the exon 5–intron 5 boundary.

#### 3.4.4. Shotgun Proteomics

MCF-7, BT-474, SK-BR3, and MDA-MB-231 cells surviving Dox treatment showed significant dysregulation of 51, 249, 316, and 32 proteins in the CYT and 2576, 414, 919, and 1498 proteins in the NE fractions, respectively. Upon Tax treatment, only five and four proteins were significantly dysregulated in the CYT fraction of MCF-7 and MDA-MB-231 cells. In the NE fractions of MCF-7, BT-474, and MDA-MB-231 cells, dysregulation of 923, 43, and 785 proteins was observed upon Tax treatment, respectively.

Shotgun proteomics did not reveal unique peptide sequences for ABCA7 in any of the NE and CYT fractions.

Although the DNA methylation status of the CpGs at the *ABCA7* exon 5–intron 5 boundary was not altered upon treatment with Dox or Tax, shotgun proteomics revealed significant down-regulation of DNA methyltransferases DNMT1, DNMT3A, and DNMT3B in MCF-7 cells surviving Dox incubation (Table 6). Significant DNMT1 down-regulation was also observed for Dox-treated SK-BR-3 and Tax-treated BT-474 cells. In addition, we found significant changes in the expression of ten–eleven translocation (TET) enzymes, playing an important role in active DNA demethylation [57], in Dox-treated MCF-7, BT-474, and SK-BR3 cells. In Dox-treated MCF-7 cells, TET3 was significantly down- and TET2 significantly up-regulated. Incubation of BT-474 and SK-BR3 cells with Dox resulted in significant down-regulation of TET2 and significant up-regulation of TET3, respectively. We also detected significant changes in the expression of methyl-CpG–binding domain (MBD) proteins, being crucial for the readout of DNA methylation [58], in all molecular subtypes after treatment with Dox. In MCF-7 cells, MBD3 and SETDB1 (SET domain-bifurcated histone lysine methyltransferase 1) were significantly down-regulated, and MBD1 and MeCP2 (methyl-CpG binding protein 2) significantly up-regulated upon treatment with Dox. In BT-474 and SK-BR3 cells, MBD proteins (MBD3; MBD2 and MBD3, respectively) were significantly down-regulated, whereas in MDA-MB-231 cells, MBD4 and MeCP2 were up-regulated. Tax treatment resulted in significant down-regulation of MBD3 in MCF-7 and MDA-MB-231 cells.

Since breast cancer cells surviving Dox or Tax treatment showed pronounced alterations in retention/alternative termination of *ABCA7* introns 3, 5, and/or 6, we used RBPmap to predict binding sites for RNA-binding proteins. Shotgun proteomics revealed significant dysregulation of a number of splicing factors predicted to bind to introns 3, 5, and/or 6 in MCF-7, SK-BR3, and MDA-MB-231 cells surviving Dox treatment, including serine/arginine-rich (SR) proteins and heterogeneous nuclear ribonucleoproteins (HNRNPs) (Table 7). SRSF2, SRSF5, and HNRNPs (with a few exceptions) were up-regulated upon treatment. RBFOX1/RBFOX2 was down-regulated in MCF-7 and SK-BR3 cells upon treatment with Dox. Tax treatment of SK-BR3 and MDA-MB-231 cells did not result in significant dysregulation of splicing factors for which binding motifs in ABCA7 introns 3, 5, and/or 6 were predicted. However, for MCF-7, EWSR1, HNRNPK, ILF2, RBM23, and RBM42 were up-regulated and ESRP1 as well as HNRNPF down-regulated upon treatment with Tax. For BT-474, neither Dox nor Tax treatment resulted in significant dysregulation of splicing factors predicted to bind to *ABCA7* introns 3, 5, and/or 6.

## 4. Discussion

We performed DNA methylation, SNP, mRNA expression, and alternative splicing analysis for *ABCA7* as well as shotgun proteomics in order to identify breast cancer-specific alterations in breast tissues, cell lines, and cell lines upon treatment with Dox or Tax.

For DNA methylation analysis, we applied PSQ because the methylation status is provided at single-nucleotide resolution in a cost-efficient manner. Previous studies on DNA methylation changes of candidate genes in cancer mainly focused on the promoter region, because alterations in promoter methylation are known to be an early and frequent event in carcinogenesis [59,60]. Promoter hypermethylation generally results in gene silencing, either by preventing binding of transcription factors or by recruiting repressive proteins with MBDs [61]. CpGs in the promoter–exon 1 region of *ABCA7* showed rather low methylation levels in breast tumor tissues, which is in accordance with TCGA data, and breast tissues from cancer-free women, indicating that the DNA methylation status of CpGs in the promoter–exon 1 region is not breast cancer-specific.

Since the target region of the respective PSQ assay contained several SNPs, we adapted the dispensation order to enable DNA methylation analysis also of DNA strands carrying minor alleles of SNPs. For the SNPs rs531508435, rs549725064, and rs3764642, automatic SNP genotyping was possible without impairing CpG methylation analysis. For rs531508435 and rs549725064, we only detected the major allele. For SNP rs3764642, we also detected the minor A allele, with the heterozygous G/A variant occurring in seven and the homozygous A variant in four out of 12 tissue samples. However, the A allele was found in both tumor and control tissues, suggesting that occurrence of the A allele of SNP rs3764642 is not breast cancer-specific. We did not find the SNP rs3764642 genotype to be associated with DNA methylation. Various ABCA7 SNPs, including rs3752246 in exon 33 [21], rs3764650 in intron 13 [22], and rs4147929 in intron 42 [23] have been identified as risk factors for AD. The clinical relevance of the rs3764642 genotype remains to be elucidated.

Growing evidence suggests that gene regulation by DNA methylation is not limited to promoter regions [62]. In particular, cancer-specific changes in gene expression were found to originate more frequently from enhancers than promoter regions [63]. In addition, several papers have reported aberrant gene body methylation, including introns in cancer, e.g., hypermethylation of *EGFR*, *LHX2*, *SOX9*, and *RFX1* introns in breast tumor tissue compared to paired adjacent tissue [64]. The first intron of eukaryotic genes is frequently longer than downstream introns [65]. With a length of 1028 bp, *ABCA7* intron 1 is actually one of the longest of the 46 introns. Intron 16 (1240 bp), 18 (1483 bp), 23 (1042 bp), 40 (2696 bp) and 42 (1230 bp) are, however, longer. *ABCA1* intron 1 contains 38 CpGs, but only two consecutive CpGs (cg10749413 and cg05989429, Figure 1) are targeted by the Infinium Human Methylation 450K BeadChip. Methylation data from TCGA indicate that in breast tumors, the two CpGs are almost unmethylated and that breast tumors from different molecular subtypes do not differ in their methylation status at these loci. Thus, we selected CpGs upstream (region A) and downstream (region B) of the two CpGs. Primer sets A and B targeted five and four of the 38 CpGs in *ABCA7* intron 1, respectively, with region B being about 480 bp downstream of region A. With the exception of CpG 2 in region A, the CpGs in intron 1 were only slightly methylated in breast tissues. The methylation status of CpG 2 was ≤23%; however, these data have to be considered with caution due to the limited applicability of PSQ to sequences containing homopolymers. Since CpG 2 follows a homopolymeric stretch of six thymines, DNA methylation levels obtained for CpG 2 were overestimated and thus excluded from statistical analyses. The methylation status of CpGs 1–3 in intron 1 targeted by primer set B was generally <LLOQ, whereas CpG 4 was slightly methylated in most tumor and normal tissue samples. Thus, neither our results nor data from TCGA hint at an association of *ABCA7* intron 1 methylation with breast cancer.

Accumulating evidence indicates that DNA methylation at exon–intron boundaries is involved in regulating alternative splicing [66], the generation of differentially spliced mRNA isoforms is from one and the same gene. Alternative splicing contributes to the enormous diversity of proteins [67]. However, aberrations in alternative splicing have been associated with a variety of diseases, including cancer [68,69]. In cancer cells, mRNA transcripts frequently show a higher incidence of exon skipping and/or intron retention events compared to normal cells [70,71,72]. To date, the mechanism of intron retention is, however, poorly understood. Intron-retaining mRNA transcripts may be subjected to nonsense-mediated decay [73,74], but they can also give rise to new protein isoforms with altered function [73]. In addition, new protein isoforms due to alternative usage of transcription initiation and/or termination are frequently found in cancer cells [75].

In general, high DNA methylation levels favor intron retention, whereas low DNA methylation levels impair binding of MeCP2, involved in recruiting splicing factors to mRNA [73]. Retained introns commonly have a higher CpG content and are shorter in length than non-retained ones [76]. With a length of 138 bp and a quantity of six CpGs, intron 5 belongs to the shortest introns with a high CpG content of the ABCA7 introns. We therefore included the exon 5–intron 5 boundary in our search for breast cancer-specific aberrations in DNA methylation and for revealing potential association with intron retention/alternative intron termination. The PSQ method based on primer set C allowed the determination of seven CpGs, with CpGs 1–4 being located in exon 5 and CpGs 5–7 in intron 5. TCGA data does not provide information on the methylation status of CpGs at the exon 5–intron 5 boundary, because the Infinium Human Methylation 450K BeadChip does not target CpGs in close vicinity (the closest CpGs upstream are cg10749413 and cg05989429 in intron 1; downstream, the closest CpG is cg23027715, located in exon 12).

In all tissue samples, the exon 5–intron 5 boundary was more highly methylated than regions A and B in intron 1. We found significant differences in the methylation status between tumor tissues of different molecular subtypes and normal breast tissues from cancer-free women. In luminal A tumors, the most prevalent breast cancer subtype, CpGs 1–7, and in luminal B tumors, CpG 1–5, were significantly more highly methylated compared to control tissues. In luminal A tumors, CpGs 1–7 and in luminal B tumors, CpG 2 and CpG 5 were significantly more highly methylated than in triple-negative tumor tissues. Previous studies have already reported that molecular subtypes of breast cancer exhibit characteristic DNA methylation patterns [77,78]. Several studies detected differences in the methylation status of the promoter of candidate genes [79,80,81,82]. Our data indicate that breast tumor tissues of different molecular subtypes may also differ in the methylation status of CpGs at exon–intron boundaries.

The presence of molecular alterations in tumor-surrounding tissues is called field cancerization or field defect [83]. There is growing evidence that cells within the epigenetic field accumulate pro-tumorigenic molecular alterations in the progression of cancer [84]. Thus, recognition of the presence of epigenetic field cancerization is deemed crucial for the early detection and prevention of breast cancer [84,85,86]. Markers for field cancerization are also important for deciphering cancer evolution [87]. In patients with luminal A breast cancer, the CpGs at the *ABCA7* exon 5–intron 5 boundary were not only more highly methylated in tumors but also in tumor-adjacent and tumor-distant tissues compared to normal breast tissues from cancer-free controls. Thus, our data suggest that for patients suffering from luminal A breast cancer, the methylation status at the *ABCA7* exon 5–intron 5 boundary is a potential biomarker for epigenetic field cancerization.

Since RNA was not available from breast tissue samples, we included sample set 2, consisting of 12 breast cancer cell lines and one non-cancerous cell line, to investigate whether DNA methylation of promoter–exon 1, intron 1 and exon 5–intron 5 boundary was correlated with *ABCA7* expression. DNA methylation levels of CpGs in promoter–exon 1 and intron 1 were in line with results obtained for breast tissues, confirming that the DNA methylation status of these regions is not breast cancer-specific. The heterozygous G/A variant and the homozygous A variant of rs3764642 were detected in seven cell lines each. The occurrence of the A allele was not breast cancer-specific. In addition, we did not find the genotype to be associated with DNA methylation.

In all breast cell lines, the exon 5–intron 5 boundary was more highly methylated than regions A and B in intron 1, with the exception of BT-474 in which the seven CpGs were largely unmethylated. In line with breast tumor tissues, the DNA methylation status depended on the molecular subtype, however, the extent of methylation was different. In Her2-positive cancer cell lines, the methylation status of the seven CpGs was significantly lower compared to the luminal A and the triple-negative subtype. In previous studies, breast cancer cell line MDA-MB-453 was classified either as Her2-positive [88,89] or triple-negative [90]. Since the DNA methylation status of MDA-MB-453 at the *ABCA7* exon 5–intron 5 boundary more closely resembled that of triple-negative than that of Her2-positive breast cancer cell lines, we suggest classifying MDA-MB-453 as triple-negative. In order to check if differences in the DNA methylation status between molecular subtypes are specific for the exon 5–intron 5 boundary or also occur in other *ABCA7* regions, several cell lines were screened for their DNA methylation status at the *ABCA7* exon 6–intron 6 boundary and in exon 12. In contrast to the exon 5–intron 5 boundary, molecular subtypes differed only slightly in the methylation status at the exon 6–intron 6 boundary and did not differ in exon 12 DNA methylation. Our findings are in line with TCGA data, indicating that molecular subtypes did not or only slightly differed in DNA methylation of exon 12, exon 13, intron 13, exon 14, exon 16, exon 18, intron 19, exon 20, intron 22, intron 29, exon 42, exon 43, exon 45, exon 46, intron 46, and downstream (<200 bp) of exon 47.

Next, we investigated if the DNA methylation levels determined were associated with *ABCA7* gene expression in the cell lines. Since various *ABCA7* transcripts have been reported in literature [24,91,92], we developed qPCR assays for the analysis of seven exon–exon boundaries (exon 1–exon 7, exon 18–19) in-house and applied them to breast cell line samples. In one and the same cell line, the nine exons were found to be expressed rather heterogeneously, but we revealed some general trends. In all cell lines, exon 1–2 boundary levels were about a factor of 2 lower than the exon 2–3 boundary. The exon 6–7 boundary was less abundant than the exon 2–3 boundary and the exon 3–4 boundary. *ABCA7* exon levels were not associated with the molecular subtype of breast cancer. For many genes, an inverse correlation between the methylation status of the first intron and gene expression has been reported [93]. We did not find *ABCA7* exon levels to be associated with DNA methylation. This holds true for all CpGs targeted.

Aberrant alternative splicing events play an important role in the onset and progression of breast cancer [70]. Due to the accumulating evidence that the DNA methylation status of CpGs at exon–intron boundaries may affect alternative splicing [66,94], we developed qPCR assays involving intron-specific primers in-house. In order to find out if *ABCA7* mRNA transcripts may contain several consecutive introns, we searched not only for intron 5 retention/alternative termination but included introns 3–4 and 6–8, two introns up- and downstream of intron 5, respectively. Interestingly, we detected retention/termination of each of the five introns in all breast cell lines investigated. Occurrence of intron retention was confirmed by applying qPCR assays using intron-flanking primer sets, followed by subjecting the PCR products to melt curve analysis and agarose gel electrophoresis. Results from both agarose gel electrophoresis and melt curve analysis revealed the presence of PCR products containing the respective intron.

In general, retention of intron 6/alternative intron 6 termination was more pronounced than intron 3–5- and intron 7-containing transcripts. Retention of intron 6/alternative intron 6 termination was associated with lower abundance of the exon 6–7 boundary. An *ABCA7* splicing variant affecting the exon 6–intron 6 boundary has already been identified by Ikeda et al. in 2003 [91]. The splicing variant has been described to affect exon 5B. However, according to NCBI exon/intron numbering, the exon termed “exon 5” is actually exon 6 and exon “5B” is intron 6. The difference in exon numbering is caused by the fact that *ABCA7* transcripts containing exon 1 were only identified by Iwamoto et al. in 2006 [9]. We did not find an association between intron retention/alternative termination and the molecular subtype of breast cancer. In addition, we did not find intron 6 retention/termination to be correlated with the DNA methylation status at the exon 5–intron 5 boundary. Since intron 6 turned out to be the most abundant of the five introns investigated, we screened seven cell lines for their DNA methylation status at the exon 6–intron 6 boundary. We did not find any association between retention/alternative termination of intron 6 and DNA methylation at the exon 6–intron 6 boundary. However, from the lack of correlation, we cannot conclude that DNA methylation at exon–intron boundaries does not have an impact on intron retention/alternative termination of intron 6. Several studies have already shown that DNA methylation in proximity to splice sites is a regulator of alternative splicing [95] and alternative transcript termination [75]. Our results only suggest that DNA methylation is not the major regulator of intron retention/alternative termination of intron 6, which is in line with the literature. Splicing factors, RNA polymerase II (RNAPII), chromatin structure and their functional associations are crucial factors affecting the splicing process [96]. Alternative transcriptional events are influenced by a combination of genetic factors, e.g., deletions and insertions, and epigenetic factors, including DNA methylation [75].

We also investigated if treatment of breast cancer cell lines with Dox or Tax has an impact on ABCA7. Dox and Tax are two important chemotherapeutic drugs for the first-line therapy of advanced breast cancer. Dox inhibits the synthesis of nucleic acid by intercalating to DNA double helix, leading to tumor cell death [97]. Tax is a microtubule poison arresting cells in mitosis [98]. It is well established that molecular subtypes of breast cancer differ in their sensitivity to chemotherapeutic substances. Thus, we selected the cell lines MCF-7, BT-474, SK-BR3, and MDA-MB-231, representing luminal A, luminal B, Her-2, and triple-negative molecular subtypes, respectively, and incubated them for 72 h with either Dox or Tax at their respective IC_50_ concentrations.

Results obtained by DNA methylation analysis revealed that neither Dox nor Tax treatment for 72 h had an impact on the DNA methylation status of at the *ABCA7* exon 5–intron 5 boundary. However, we observed altered *ABCA7* transcript levels upon treatment. By applying qPCR assays involving exon-specific primers, *ABCA7* was found to be up-regulated in MCF-7 and down-regulated in BT-474 cells upon Dox treatment. In Dox-treated SK-BR3 cells, only the exon 18–19 boundary was up-regulated, whereas in MDA-MB-231 cells, Dox did not have any impact on *ABCA7* exon–exon boundary levels. Compared to Dox, the effect of Tax on *ABCA7* expression was modest. We only observed a reduction in the expression of the exon 18–19 boundary in SK-BR3 cells.

We detected alternative splicing events in *ABCA7* upon treatment. In MCF-7, SK-BR3, and MDA-MB-231 cells surviving Dox treatment, we found increased intron 3 and intron 6, intron 3 and intron 5, and intron 6 levels, respectively. In Tax-treated MCF-7 cells, intron 5 was less abundant, whereas Tax treatment of SK-BR3 was associated with increased intron 5 retention/alternative termination compared to untreated cells.

In order to reveal global changes in protein expression upon Dox or Tax treatment, CYT and NE fractions of the cells surviving treatment were subjected to shotgun proteomics. Dox resulted in a higher number of dysregulated proteins than Tax, and as expected, more dysregulated proteins were found in NE fractions than in the respective CYT fractions. Unfortunately, ABCA7 could not be detected in any of the fractions analyzed. Due to the hydrophobic nature of membrane-embedded domains, shotgun proteomics of integral membrane proteins such as ABCA7 is challenging. Due to 54% sequence identity with ABCA1 [4], identification of ABCA7 is additionally hampered.

Although we did not observe alterations in DNA methylation at the *ABCA7* exon 5–intron 5 boundary, shotgun proteomics revealed significant down-regulation of the de novo DNA methyltransferases DNMT3A and DNMT3B as well as the maintenance DNA methyltransferase DNMT1. In MCF-7 cells treated with Dox, all three DNA methyltransferases were down-regulated. Down-regulation of DNMT1 was also observed for SK-BR3 cells upon Dox and in BT-474 cells upon Tax treatment. DNMT1 is known to be one of the major targets of Dox, as it inhibits its enzymatic activity in vitro via DNA intercalation [99]. Growing evidence suggests that inhibition of DNMTs could enhance chemotherapeutic sensitivity of tumor cells. Thus, a combined therapy of DNMT inhibitors with chemotherapeutic drugs might be an attractive approach to circumvent drug resistance [100]. In addition to alterations in the expression of DNA methyltransferases, we found significant changes in the expression of TET enzymes and MBD proteins. TET proteins, including TET1, TET2, and TET3, are crucial for active DNA demethylation by catalyzing oxidation of 5-methylcytosine to 5-hydroxymethylcytosine [101]. Members of the MBD protein family are DNA methylation readers. They recruit chromatin remodelers, histone deacetylases, and methylases to methylated DNA, leading to transcriptional repression [102]. In contrast to DNA methyltransferases, we did not find a clear tendency for up- or downregulation of TET and/or MBD proteins upon Dox or Tax treatment, with a few exceptions. MeCP2 was significantly up-regulated in MCF-7 and MDA-MB-231 cells upon Dox treatment. MeCP2 expression has already been associated with breast cancer in a previous study, reporting MeCP2 mRNA expression to be strongly associated with the estrogen receptor status [103]. MBD3 was significantly down-regulated in MCF-7, BT-474, and SK-BR3 cells upon Dox and MCF-7 and MDA-MB-231 cells upon Tax treatment. Growing evidence suggests that MBD3 lacks specificity toward 5-methylcytosine but binds 5-hydroxymethylcytosine [102].

Since we observed pronounced intron retention/termination in *ABCA7* mRNA transcripts upon treatment with Dox or Tax by qPCR involving intron-specific primers, we were interested in whether aberrant splicing is associated with dysregulation of proteins involved in RNA processing. The spliceosome is, however, a large complex, consisting of five small nuclear RNAs (snRNAs) and hundreds of proteins [104]. There is accumulating evidence that multiple factors determine the final outcome of the splicing reaction, including strength and context of binding sites on pre-mRNA; expression levels, localization and mutations of individual RNA-binding proteins, as well as the ratio of positive and negative regulatory splicing factors [70,94,105]. Due to the complexity, we primarily searched shotgun proteomics data for splicing factors for which binding motifs in *ABCA7* introns 3, 5, and/or 6 were predicted. Shotgun proteomics revealed that a high number of splicing factors fulfilling this criterion were dysregulated. Dysregulation of splicing proteins was observed for MCF-7, SK-BR3, and MDA-MB-231 cells upon treatment with Dox and MCF-7 cells upon treatment with Tax. For BT-474, drug treatment had no influence on *ABCA7* introns retention/alternative termination compared to untreated cells. In line with these results, shotgun proteomics did not reveal significant dysregulation of splicing factors predicted to bind to *ABCA7* introns 3, 5, and/or 6 in treated BT-474 cells.

Pronounced detection of *ABCA7* introns 3 and 6 in Dox-treated MCF-7 cells was associated with down-regulation of several splicing factors, including CELF1, ESRP1, FUBP1, PCBP1, PCBP2, PUF60, RBFOX1/RBFOX2, and RBM24/RBM38. Other splicing factors, including several HNRNPs, EWSR1, ILF2, and SRSF5, were found to be up-regulated. In Dox-treated SK-BR3 cells, pronounced detection of *ABCA7* intron 3 was associated with down-regulation of RBFOX1/RBFOX2, RBM24/RBM38, ESRP2, PUM1, and ZC3H14, whereas ESRP1, FUS, HNRNPM, and SRSF2 were up-regulated. Analysis of the NE fraction of Dox-treated MDA-MB-231 cells, exhibiting increased intron 6 levels, with the exception of CELF1 splicing factors predicted to bind to introns 3, 5, and/or 6 were up-regulated, including HNRNPA0, RBM6, RBM23, SRSF2, and SRSF5. Upon treatment with Tax, MCF-7 cells exhibited decreased *ABCA7* intron 5 levels. Increased *ABCA7* intron 5 levels were associated with up-regulation of EWSR1, HNRNPK, ILF2, RBM23, and RBM42, whereas ESRP1 and HNRNPF were down-regulated.

Our results indicate that intron retention and/or alternative intron termination, both resulting in non-functional ABCA7, is an important mechanism for cancer cells to cope with chemotherapy-induced stress. Reduced splicing in cancer cells due to chemotherapy-induced stress response has already been reported recently by Anufrieva et al. [106]. They detected global changes in pre-mRNA splicing in cancer cells upon treatment with a variety of chemotherapeutic drugs. Moreover, different chemotherapeutic drugs were found to lead to similar changes by inducing intron retention in multiple genes [106].

## 5. Conclusions

There is accumulating evidence that breast cancer is linked to dysregulation of cholesterol homeostasis. The ABC transporter ABCA7, involved in lipid transport processes and cholesterol homeostasis, has been found to be aberrantly expressed in a variety of cancer types, including breast cancer. By searching for breast cancer-specific alterations, we found promoter–exon 1 and intron 1 regions unmethylated in tumor tissues from breast cancer patients and breast tissues from cancer-free women. However, we detected breast cancer-specific aberrant DNA methylation at the exon 5–intron 5 boundary of *ABCA7*. In patients suffering from luminal A breast cancer, aberrant DNA methylation was also detected in tumor-adjacent tissues appearing histologically normal. Thus, the DNA methylation status at the exon 5–intron 5 boundary of *ABCA7* could serve as a potential marker for epigenetic field cancerization.

In commercial breast cancer cell lines, DNA methylation of promoter–exon 1, intron 1, and the exon 5–intron 5 boundary was not correlated with *ABCA7* gene expression. However, analysis of *ABCA7* mRNA transcripts revealed differences in the expression of *ABCA7* exons. By qPCR involving intron-specific primers, we detected *ABCA7* intron retention/alternative termination in a molecular subtype-specific manner. Although accumulating evidence suggests that the DNA methylation status at exon–intron boundaries has an impact on alternative splicing, we did not find *ABCA7* intron retention/alternative termination to be directly correlated with DNA methylation, suggesting that DNA methylation is not the major regulator.

Treatment of four breast cancer cell lines with Dox or Tax for 72 h resulted in altered retention of *ABCA7* introns in a molecular subtype-specific manner. Shotgun proteomics revealed that aberrant intron retention/alternative intron termination was associated with significant dysregulation of splicing factors linked to alternative splicing. Significant dysregulation upon treatment with Dox or Tax affected numerous other proteins, e.g., DNA methyltransferases, TET enzymes, and MBD proteins. However, unique peptide sequences were not obtained for ABCA7, most probably due to the hydrophobic nature of membrane-embedded domains and high sequence identity with ABCA1.

One limitation of our study is that we applied qPCR using intron-specific primers to identify alternative splicing variants in *ABCA7*, which cannot distinguish between intron retention and alternative intron termination. In addition, our study did not cover all transcript variants of *ABCA7*, comprising 47 exons. The whole *ABCA7* transcriptome and its regulation in breast cancer remains to be investigated in further studies, e.g., by RNA sequencing.

## Figures and Tables

**Figure 1 cells-12-01462-f001:**
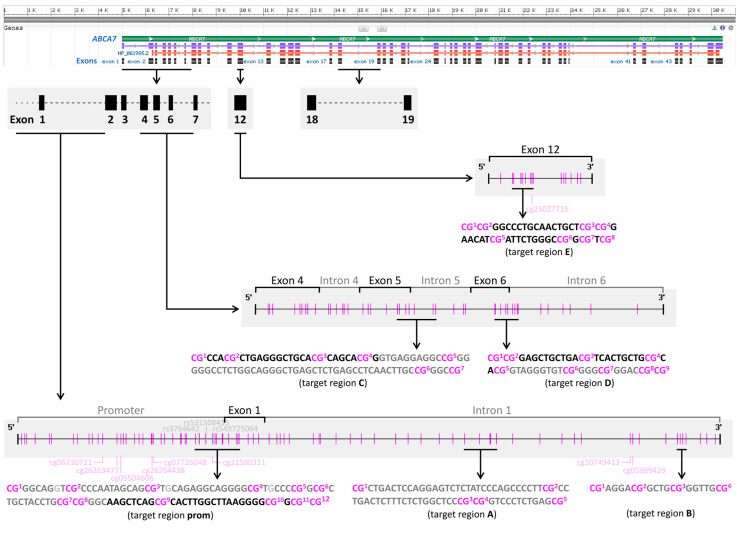
Schematic representation of the regions of the *ABCA7* gene targeted by DNA methylation analysis. Promoter region according to Iwamoto et al. [9] (gray dotted line upstream of exon 1), exons (black bars), and introns (gray dashed lines) are shown. Pink vertical lines indicate CpG positions, light-gray vertical lines SNPs (rs3764642, rs531508435, and rs549725064). In the sequences to analyze covering coding (black) and non-coding (gray) regions, the CpGs are numbered according to their order in the respective pyrogram. Nine CpGs (cg06730721 to cg23027715) are covered by the Infinium Human Methylation 450K BeadChip. CpG positions were visualized with the help of Methyl Primer Express software v1.0 (Thermo Fisher Scientific, Waltham, MA, USA).

**Figure 2 cells-12-01462-f002:**
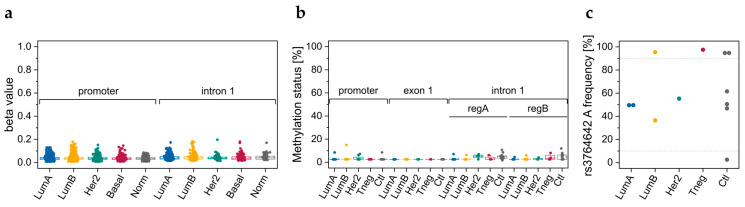
DNA methylation status of the *ABCA7* promoter–exon 1 and intron 1 region and SNP rs3764642 genotypes in breast tissue samples. (**a**) Beta values for six CpGs in the promoter region (cg06730721–cg21590311) and intron 1 (cg10749413, cg05989429) for 372 luminal A, 125 luminal B, 44 Her2-positive, 112 basal-like, and 30 normal-like molecular subtypes of 683 TCGA breast cancer tissues samples. (**b**) Methylation status of eight CpGs in the promoter region, four CpGs in exon 1, and four CpGs in region A and region B of intron 1, respectively, in six tumor and four control tissue samples. (**c**) SNP rs3764642 A frequency for six tumor and six control tissue samples. No significant differences were observed in the methylation levels or genotypes between molecular subtypes.

**Figure 3 cells-12-01462-f003:**
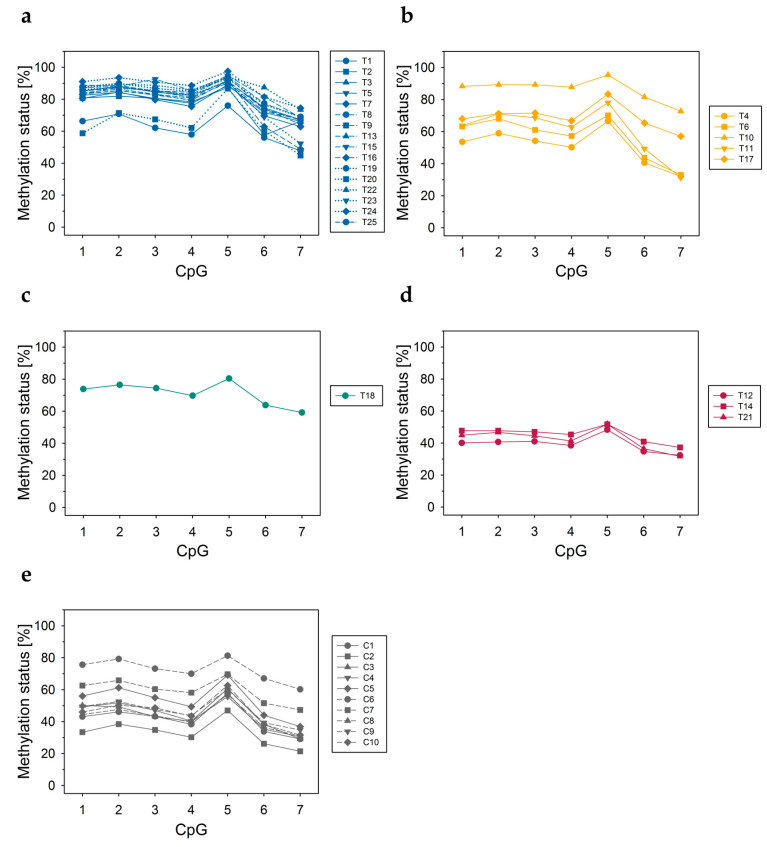
DNA methylation status of CpGs 1–7 at the *ABCA7* exon 5–intron 5 boundary for tumor tissues from (**a**–**d**) 25 breast cancer patients and (**e**) 10 normal breast tissues from cancer-free women. (**a**) Luminal A, (**b**) luminal B, (**c**) Her2-positive and (**d**) triple-negative. T: tumor, C: control. The sample ID number of tumor tissues refers to the patient number listed in Table 1. Each data point shows the arithmetic mean of at least two technical replicates.

**Figure 4 cells-12-01462-f004:**
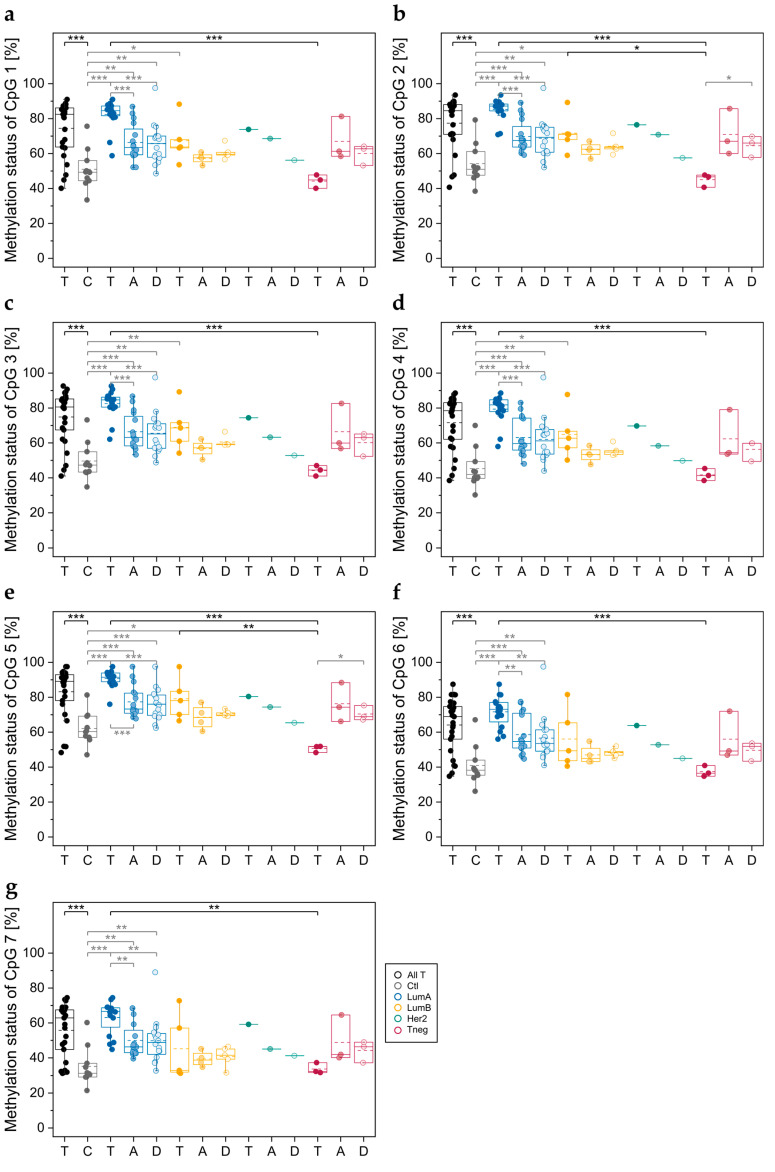
Distribution of the DNA methylation status of CpGs 1–7 at the *ABCA7* exon 5–intron 5 boundary for tumor tissues from 25 breast cancer patients and 10 normal breast tissues from cancer-free women. (**a**) CpG 1, (**b**) CpG 2, (**c**) CpG 3, (**d**) CpG 4, (**e**) CpG 5, (**f**) CpG 6, and (**g**) CpG 7. All T: tumor tissue samples from all molecular subtypes LumA: luminal A; LumB: luminal B; Her2: Her2-positive; Tneg: triple-negative; C: controls. T: tumor center, A: adjacent, D: distant tissues. Each data point shows the arithmetic mean of at least two technical replicates. Solid line: median, dashed line: arithmetic mean. Significance by molecular subtypes (black) and by field cancerization effects (gray). * *p* < 0.05, ** *p* < 0.01, *** *p* < 0.001.

**Figure 5 cells-12-01462-f005:**
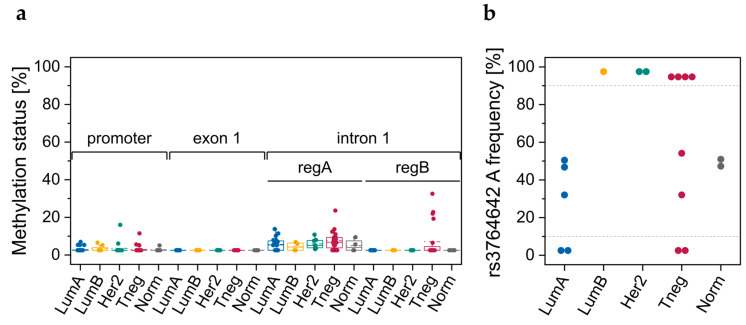
DNA methylation status of the *ABCA7* promoter–exon 1 and intron 1 region and SNP rs3764642 genotypes in commercial breast cell lines. (**a**) Methylation status of eight CpGs in the promoter region, four CpGs in exon 1, and four CpGs in region A and region B of intron 1, respectively, in 13 breast cancer cell lines and one non-cancerous breast cell line. (**b**) SNP rs3764642 A frequency for 16 breast cancer cell lines and two non-cancerous cell lines. No significant differences were observed in the methylation levels or genotypes between molecular subtypes.

**Figure 6 cells-12-01462-f006:**
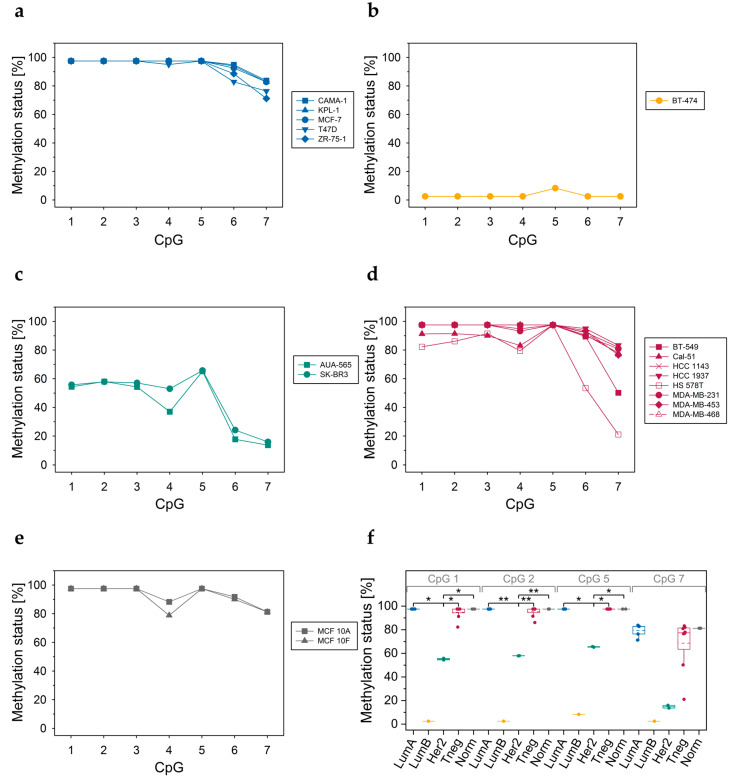
DNA methylation status of CpGs 1–7 at the *ABCA7* exon 5–intron 5 boundary for (**a**–**d**) 16 breast cancer and (**e**) two non-cancerous breast cell lines. (**a**) Luminal A, (**b**) luminal B, (**c**) Her2-positive, (**d**) triple-negative cell lines. (**f**) Distribution of the methylation status of CpGs 1, 2, 5; and 7. LumA: luminal A; LumB: luminal B; Her2: Her2-positive; Tneg: triple-negative; norm: non-cancerous breast cell lines. Each data point shows the arithmetic mean of at least two technical replicates. (**f**) Solid line: median, dashed line: arithmetic mean. Significance by molecular subtypes (black). * *p* < 0.05, ** *p* < 0.01.

**Figure 7 cells-12-01462-f007:**
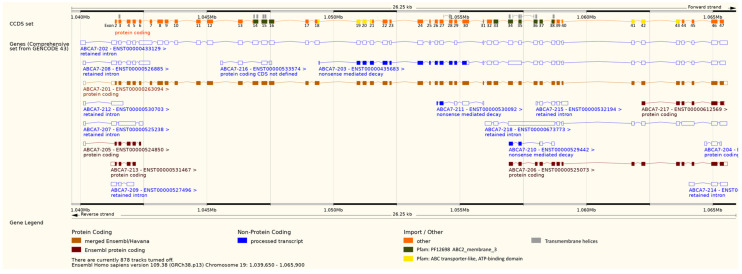
Overview of GENCODE 43 annotated *ABCA7* transcript variants taken from the Ensembl Genome Browser [56]. The coding sequence for the CCDS protein was annotated manually to visualize exons encoding transmembrane domains (dark green), exons encoding ATP-binding domains (yellow), and other coding exons (orange). In addition, transmembrane helices (gray) are shown above the exons.

**Figure 8 cells-12-01462-f008:**
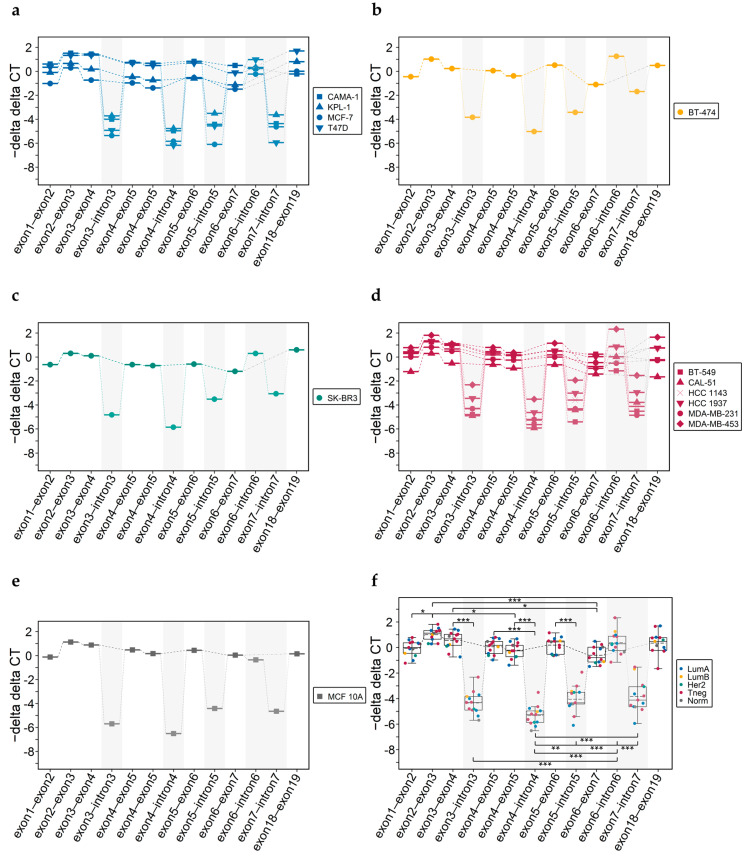
*ABCA7* transcript levels obtained by using exon-specific (white fill) and intron-specific (light-gray fill) primers. (**a**) Luminal A, (**b**) luminal B, (**c**) Her2-positive, (**d**) triple-negative breast cancer cell lines. (**e**) non-cancerous breast cell line, (**f**) all cell lines. LumA: luminal A; LumB: luminal B; Her2: Her2-positive; Tneg: triple-negative; norm: non-cancerous breast cell lines. Each data point shows the arithmetic mean of at least two technical replicates. (**f**) Solid line: median, dashed line: arithmetic mean. Significance between exon–exon boundaries, introns, and exons to adjacent intron (black). * *p* < 0.05, ** *p* < 0.01, *** *p* < 0.001. No significant difference was found between molecular subtypes. ΔΔCT refers to MDA-MB-231 exon 5–exon 6 to allow comparison between different assays.

**Figure 9 cells-12-01462-f009:**
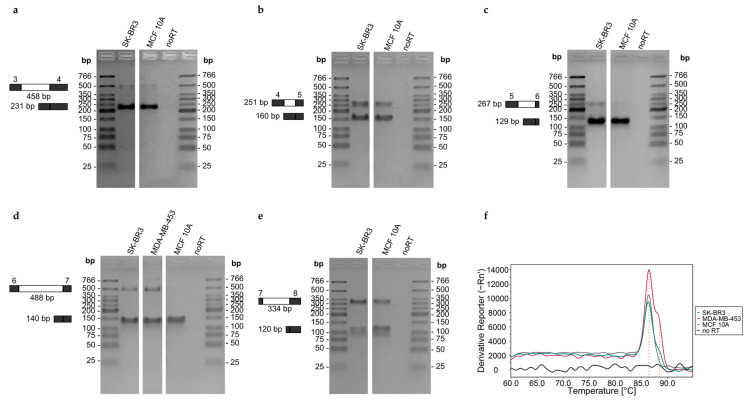
Verification of *ABCA7* intron retention by (**a**–**e**) agarose gel electrophoresis (shown for selected samples; results for all samples are shown in Appendix A) and (**f**) melt curve analysis. Intron-flanking primer sets: (**a**) exon 3–(intron 3)–exon 4, (**b**) exon 4–(intron 4)–exon 5, (**c**) exon 5–(intron 5)–exon 6, (**d**,**f**) exon 6–(intron 6)–exon 7, and (**e**) exon 7–(intron 7)–exon 8. Schemes show exons (dark-gray fill), introns (white fill), and the expected PCR product length.

**Figure 10 cells-12-01462-f010:**
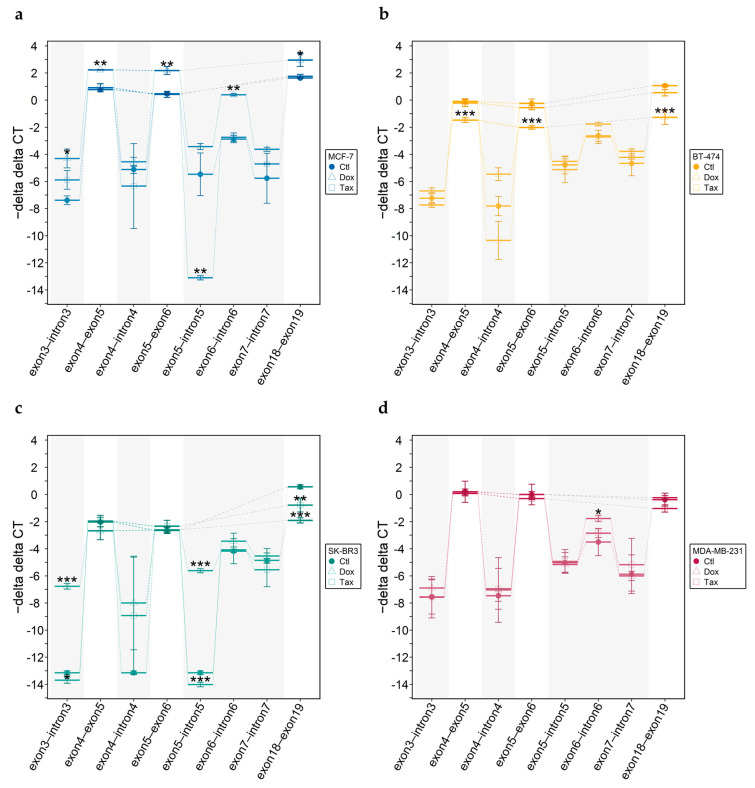
*ABCA7* transcript levels obtained by exon-specific (white fill) and intron-specific (light-gray fill) primers. (**a**) Luminal A, (**b**) luminal B, (**c**) Her2-positive, (**d**) triple-negative cell lines. Dox: doxorubicin, tax: paclitaxel. Solid line: arithmetic mean. ΔΔCT refers to untreated MDA-MB-231 exon 5–exon 6 to allow comparison between different assays. Significance by treatment versus control (black) * *p* < 0.05, ** *p* < 0.01, *** *p* < 0.001.

**Table 1 cells-12-01462-t001:** Patient and tumor characteristics.

Patient	Age [y]	Menopause	Histological	Histological	MIB-1 [%]	Receptor Status			Molecular
		Status	Type	Grading		ER	PR	Her2	Subtype
1	75	Post	IDC	2	10	+	+	−	Luminal A
2	65	Post	IDC	2	10	+	+	−	Luminal A
3	54	Peri	IDC	3	30	+	−	−	Luminal A
4	39	Pre	IDC	2	40	+	+	+	Luminal B
5	66	Post	IDC	2	60	+	+	−	Luminal A
6	50	Pre	IDC	3	50	+	+	+	Luminal B
7	73	Post	IDC	3	20	+	+	−	Luminal A
8	76	Post	IDC	2	20	+	+	−	Luminal A
9	63	Post	IDC	3	30	+	+	−	Luminal A
10	48	Post	IDC	3	20	+	+	+	Luminal B
11	58	Post	IDC	1	20	+	+	+	Luminal B
12	61	Post	IDC	3	70	−	−	−	Triple-negative
13	52	Pre	ILC	2	50	+	+	−	Luminal A
14	42	Pre	IDC	3	80	+/−	−	−	Triple-negative ^1^
15	67	Post	IDC	3	40	+	+	−	Luminal A
16	61	Post	ILC	2	30	+	+	−	Luminal A
17	41	Pre	Mucinous	2	50	+	+	+	Luminal B
18	59	Post	IDC	3	50	−	−	+	Her2-positive
19	82	Post	IDC	3	30	+	−	−	Luminal A
20	86	Post	IDC	3	40	+	+	−	Luminal A
21	43	Pre	IDC	3	30	−	−	−	Triple-negative
22	50	Pre	IDC	3	60	+	+	−	Luminal A
23	55	Post	IDC	3	10	+	+	−	Luminal A
24	62	Post	IDC	3	70	+	−	−	Luminal A
25	32	Pre	IDC	2	30	+	+	−	Luminal A

^1^ The tumor showed only moderate expression (50%) of the estrogen receptor, and in consequence of chemotherapeutic treatment, expression of the estrogen receptor decreased to 10%. In a previous study, the methylation pattern corresponded to that of a triple-negative tumor [40]. (MIB-1: Ki-67 labeling index, determined by using MIB-1 antibodies, IDC: invasive ductal carcinoma, ILC: invasive lobular carcinoma, ER: estrogen receptor, PR: progesterone receptor, Her2: human epidermal growth factor receptor 2, +: positive, −: negative).

**Table 2 cells-12-01462-t002:** DNA methylation analysis of *ABCA7* by PSQ.

Primer Set	Primer Sequence (5′→3′)	Amplicon Length [bp]	Number of CpGs Analyzed	Primer Conc. [nM]	T_a_ [°C]
Prom	F: GTTTYGAYGGAGTAGGTTAGTGA	156	12	200	58.5
	R: [Btn]CTTCCCTCCTCCCAACAACA				
	S: GAGTAGGTTAGTGAG				
A	F: GGTGGATAGGTTTAAGGAAAATAGA	138	5	400	57.0
	R: [Btn]CCAACTCTACCCACACCTTATA				
	S: GGTTTAAGGAAAATAGA				
B	F: AGATAGTTTTGGTAGTTAATTAGATG	78	4	400	57.0
	R: [Btn]CTACCCCAAAAATAAAATAAACCAATA				
	S: AGTTTTGGTAGTTAATTAGATG				
C	F: TTGGTTGGTTTAGGGAAGTTG	131	7	200	58.3
	R: [Btn]CCCACCCTATACCCATTTC				
	S: GGTTTAGGGAAGTTGAT				
D	F: TTAATTAAGTAGTTTTTATTGGAATTAT	130	9	400	51.0
	R: [Btn]AAAACATCCCCAATACAATATC				
	S: TTTATTGGAATTATTTATGT				
E	F: TAGTGTTTGTTTTTGGATAAGTTGGA	164	8	200	55.4
	R: [Btn]AAACCCAAATCTAAAATTAAATACTCTATA				
	S: GGTATTTTTAGAGGTAG				

[Btn]: biotin, bp: base pairs, conc.: concentration, F: forward primer, R: reverse primer, S: sequencing primer, T_a_: annealing temperature, T: substitute base for R.

**Table 3 cells-12-01462-t003:** Exon-specific primer sets for gene expression analysis.

Gene	Primer Set	Primer Sequence (5′→3′)	Location	Amplicon Length [bp]
*ABCA7*	exon 1–exon 2	F: GGAGCCCCTGGAAGAGC	boundary	127
		R: CAGAGAGGTCAGGACAACG	exon 2	
	exon 2–exon 3	F: ACCGTTGTCCTGACCTCTC	exon 2	118
		R: GCTGGACCGGCTGTCTC	boundary	
	exon 3–exon 4	F: CACCATGAATGCCACTTCC	boundary	150
		R: GGAGTCGTTGAAGTTGCTCA	exon 4	
	exon 4–exon 5 set 1	F: CAACTTCAACGACTCCCTGGT	boundary	82
		R: CAGCGTCCTGTGGGCACT	exon 5	
	exon 4–exon 5 set 2	F: GCTCCAGGGTCTCATCTGTA	exon 4	103
		R: GGAGACCAGGGAGTCGTTG	boundary	
	exon 5–exon 6	F: TAGCCGATGCCCGCACTG	exon 5	109
		R: TGAGGCTGGGCCGTGCT	boundary	
	exon 6–exon 7	F: ACCAACCAAGCAGTCTCCAC	exon 6	88
		R: CAACCCCAGGGATTCCGTG	boundary	
	exon 18–exon 19	F: ACCACCACCCTGTCCATC	boundary	137
		R: AACAGCACGTTGTACTGAGG	exon 19	
*PUM1*	housekeeping gene	F: TACCGCTCCGCCCGTGT	boundary	130
	(exon 1–exon 2)	R: CAGGCATGTTGGGATTAGCTG	exon 2	

bp: base pairs, F: forward primer, R: reverse primer.

**Table 4 cells-12-01462-t004:** Intron-specific primer sets for analysis of intron-containing transcripts.

Gene	Primer Set	Primer Sequence (5′→3′)	Location	Amplicon Length [bp]
*ABCA7*	exon 3–intron 3	F: CAGCTCCTGGTCGAATTGC	exon 3	192
		R: CTGGTGAAGGCTCCCTGA	intron 3	
	exon 4–intron 4	F: GCTCCAGGGTCTCATCTGTA	exon 4	119
		R: CGCACCCACTGCCTCTG	intron 4	
	exon 5–intron 5	F: TAGCCGATGCCCGCACTG	exon 5	176
		R: CACCCTGTGCCCATTTCAC	intron 5	
	exon 6–intron 6	F: ACCAACCAAGCAGTCTCCAC	exon 6	218
		R: CAGACGGACAAGCAGCATCA	intron 6	
	exon 7–intron 7	F: CCTTGCACAGCTTGTTGGAG	exon 7	168
		R: CTCGCTGTCTAATCCTCCTG	intron 7	

bp: base pairs, F: forward primer, R: reverse primer.

**Table 5 cells-12-01462-t005:** Intron-flanking primer sets for verification of intron retention.

Gene	Primer Set	Primer Sequence (5′→3′)	Location	Amplicon Length ^1^ [bp]
*ABCA7*	exon 3–intron 3–exon 4	F: CAGCTCCTGGTCGAATTGC	exon 3	458/231
		R: GGAGTCGTTGAAGTTGCTCA	exon 4	
	exon 4–intron 4–exon 5	F: GCTCCAGGGTCTCATCTGTA	exon 4	251/160
		R: CAGCGTCCTGTGGGCACT	exon 5	
	exon 5–intron 5–exon 6	F: TAGCCGATGCCCGCACTG	exon 5	267/129
		R: GTGGAGACTGCTTGGTTGGT	exon 6	
	exon 6–intron 6–exon 7	F: ACCAACCAAGCAGTCTCCAC	exon 6	488/140
		R: CCTCAGCGGCCTCCAACA	exon 7	
	exon 7–intron 7–exon 8	F: TGTTGGAGGCCGCTGAGG	exon 7	334/120
		R: CCTCTGACAGCAACTCCAG	exon 8	

^1^ Amplicon length with and without intron retention, respectively. (bp: base pairs, F: forward primer, R: reverse primer).

**Table 6 cells-12-01462-t006:** Significantly dysregulated DNA methyltransferases, ten–eleven translocation (TET) enzymes, and methyl-CpG–binding domain (MBD) proteins upon treatment.

Cell Line	Treatment	Fraction	Protein	Fold Change (Log 2)	*p* Value ^1^
MCF-7	Dox	CYT	TET3	−4.89	<0.001
		NE	DNMT1	−2.19	<0.01
			DNMT3A	−1.12	<0.01
			DNMT3B	−2.36	<0.05
			TET2	3.53	<0.01
			TET3	−1.54	<0.01
			MeCP2	1.20	<0.05
			MBD1	4.34	<0.05
			MBD3	−3.15	<0.001
			SETDB1	−2.53	<0.05
	Tax	NE	MBD3	−0.63	<0.01
BT-474	Dox	NE	TET2	−2.80	<0.001
			MBD3	−2.35	<0.01
	Tax	NE	DNMT1	−1.32	<0.001
SK-BR3	Dox	CYT	TET3	6.48	<0.001
		NE	DNMT1	−3.48	<0.01
			MBD2	−2.05	<0.01
			MBD3	−2.16	<0.001
MDA-MB-231	Dox	NE	MeCP2	2.61	<0.05
			MBD4	3.65	<0.05
	Tax	NE	MBD3	−1.92	<0.01

^1^ Multiple testing corrected using false-discovery rate (FDR = 0.05).

**Table 7 cells-12-01462-t007:** Significantly dysregulated splicing factors predicted to bind to *ABCA7* intron 3, intron 5, and/or intron 6.

Cell Line	Treatment	Fraction	Protein	Fold Change (Log 2)	*p* Value ^1^	Predicted Binding Site		
						Intron 3	Intron 5	Intron 6
MCF-7	Dox	NE	CELF1	−2.36	<0.001	+	−	+
			ESRP1	−2.09	<0.001	+	+	−
			EWSR1	1.44	<0.01	+	+	−
			FUBP1	−4.67	<0.001	+	−	−
			FXR2	2.30	<0.01	−	−	+
			HNRNPA0	1.41	<0.01	−	−	+
			HNRNPA2B1	1.10	<0.001	+	−	+
			HNRNPF	−0.57	<0.01	+	+	+
			HNRNPH2	0.82	<0.05	+	+	+
			HNRNPU	1.49	<0.001	−	−	+
			ILF2	1.64	<0.001	+	+	+
			KHSRP	−2.88	<0.01	+	−	−
			PCBP1	−1.14	<0.01	+	−	+
			PCBP2	−0.93	<0.05	+	−	+
			PUF60	−4.10	<0.05	−	−	+
			RBFOX1/ RBFOX2	−1.29	<0.05	+	−	−
			RBM8A	0.71	<0.05	−	−	+
			RBM24/ RBM38	−3.49	<0.01	+	+	+
			SNRNP70	−0.45	<0.01	−	−	−
			SRSF5	0.62	<0.01	+	−	+
			TAF15	1.08	<0.05	+	−	+
			TARDBP	−1.18	<0.01	+	−	+
			ZC3H14	−1.77	<0.001	+	−	−
	Tax	NE	ESRP1	−0.88	<0.001	+	+	−
			EWSR1	1.04	<0.01	+	+	−
			HNRNPF	−0.90	<0.001	+	+	+
			HNRNPK	0.64	<0.01	−	+	−
			ILF2	1.07	<0.01	+	+	+
			RBM23	1.84	<0.05	−	+	+
			RBM42	1.01	<0.01	−	+	−
SK-BR3	Dox	CYT	HNRNPH2	1.29	<0.01	+	+	+
		NE	ESRP1	2.05	<0.05	+	+	−
			ESRP2	−4.71	<0.001	−	+	−
			FUS	4.55	<0.001	−	+	+
			HNRNPM	1.25	<0.05	−	+	−
			PUM1	−2.74	<0.01	+	−	−
			RBFOX1/ RBFOX2	−2.24	<0.05	+	−	−
			RBM24/ RBM38	−2.65	<0.001	+	+	+
			SRSF2	1.06	<0.05	+	+	+
			ZC3H14	−4.81	<0.001	+	−	−
MDA-MB-231	Dox	NE	CELF1	−1.80	<0.05	+	−	+
			FUS	0.70	<0.05	−	+	+
			FXR2	1.06	<0.001	−	−	+
			HNRNPA0	0.48	<0.01	−	−	+
			RBM6	0.51	<0.01	−	−	+
			RBM23	1.77	<0.01	−	+	+
			RBM25	0.89	<0.05	−	−	+
			SRSF2	1.52	<0.05	+	+	+
			SRSF5	0.61	<0.01	+	−	+
			TAF15	1.15	<0.05	+	−	+
			ZNF326	0.93	<0.05	−	−	+

^1^ Multiple testing corrected using false-discovery rate (FDR = 0.05). (+: present, −: absent).

## Data Availability

The datasets generated during the current study are available from the corresponding author on reasonable request.

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
