# Peer review of "Aberrant DNA Methylation, Expression, and Occurrence of Transcript Variants of the ABC Transporter ABCA7 in Breast Cancer"

_cells, 2023, doi:10.3390/cells12111462_

Round 1
Reviewer 1 Report
Zappe et al. address the implications of ABCA7, a type of cell membrane transporter in the various hallmarks of breast cancer. However, the experimental approaches in addressing above proposition is elaborative and sufficient.
Some suggestions may help in the better impact of this paper.
1. Authors are suggested to bring better rationale to connect the role of ABCA7 with the molecular pathogenesis of breast cancer.
2. The authors are recommended to show the expressional level of ABCA7 in these breast cancer cells and tissues employing immunoblotting or ICC.
3. A perspective on ABCA7 and cancer associated adipocytes may be discussed.
4. The authors may revisit title of manuscript, since title does not fully make coherence with the manuscript.
5. The authors should make clear the role of ABCA7 as an oncoprotein or tumor suppressors. Since, due to complex nature of cancer, various proteins play a dual role.
Reviewer 2 Report
In the last paragraphs of the Introduction, the authors describe how the groups were formed and what tests were made to the obtained samples and cell lines. This information should be added only to the Materials and Methods section. As it is, it feels repetitive. If the authors feel the need to flesh out the performed procedures, they should do so in the appropriate section. The introduction should contain only the necessary information to “set the scene for the manuscript”, ending with a brief summary of the main goals.
In table 5, the authors show the ABC transporters that had their expression disrupted one way of another by the treatment. It's worth noting that none of the three most studied transporters within the context of cancer, namely ABCC1, ABCB1 and ABCG2 were apparently not altered. There are many works in which those proteins are reportedly altered, using many different cell lines treated withg both doxorubicin and paclitaxel (10.1016/j.matbio.2023.03.002, 10.1016/j.taap.2016.09.020). It would be interesting to add a paragraph or two to the discussion and cover that.
Reviewer 3 Report
Aberrant DNA methylation and expression of the ABC transporter ABCA7 in breast cancer
By Katja Zappe et al. 2023
The aim of this study was to determine whether a correlation could exist between the global RNA level and alternative splicing isoforms expressed by the ABCA7 gene in breast cancers with the level of intragenic methylation at CpG in exon5 and in exon1 regions. ABC7 gene product was involved in the lipid and cholesterol transport/homeostasis and a role of ABCA7 was suspected in cancers by few studies : in particular low level of ABCA7 mRNAs in colorectal cancer is of bad prognostic.
The part of the study describing the variation of DNA methylation in breast cancers or in cell lines is well conducted (fig 2 to 6), even one or more "CpG reference" could be added to show whether the variation of the ABCA7 loci where significative of a global change or a specific change. The part of the study (fig 7 & 8) of alternative splicing is more weak because it has been conducted only by quantitative PCR with very short amplicons without trying to test whether the hypothetic design of primers could take in account the real transcriptional variations. In particular alternative promoter or termination were not at all considered. A third part of the study consists to the proteomic analysis of cell lines after chemotherapy, but weakly linked to the rest of the paper since the ABCA7 proteins were not detectable. The variation of the full list of splicing factors hypothetically regulators of ABCA7 were finally shortly discussed
I cannot recommand publication of this study without adding several controls, in particular on the RNA study part : because what the author named "intron retention" could be several other (also) interesting things. I suggest few experiments to study further the consequence of DNA methylation variation on ABCA7 protein expression which was not possible by mass-spec. Finally maybe I'd like suggest a reflexion about this field.
Detailled remarks :
The authors study the modification of DNA methylation in breast cancers compared to normal or neighbor tissues within five different regions in the 5' part of the ABCA7 gene. It is difficult to judge from these data whether the observed modifications originated from local change in the ABCA7 gene or from a more global change as a general decrease of DNA methylation. Maybe few well-known CpG rich regions (p16 promoter ? other already described?) used as reference would be usefull to compare with those in ABCA7.
The authors want to study " the formation of ABCA7 splicing variants " (line 696), but their analysis of exon-exon bundaries was very limited since they only document consecutive exon-exon junctions and never potential alternative cassette exon, nor alternative splice junction. So this quantitative analysis of exon level poorly described the potential alternative "splicing variants" of ABCA7.
For instance, the authors found (line 862) that " In all cell lines, exon 1 levels were 862 about a factor of 2 lower than levels of exons 2–4.". This observation is probably biased because of the experimental design. Indeed the "exon1" level is measured by RT-qPCR using exon1-exon2-forward primer (e1e2F) and exon2 reverse primer (e2R) (table3). This prevents the authors to measure the level of exon1 bound to the exon2 by at least two different alternative junction described in database (ENST00000263094 or ENST00000524850). Also, of course, database maybe incomplete ; so the only way to be sure the exon1 level is correctly evaluated (without considering deep-sequencing assays) is to be performed PCR with primers not on junction : the PCR with e1F +e2R followed by an electrophoresis of PCR product will indicate the main transcripts expressed by the different cell lines in different treatments. Sanger sequencing of these product will allow to design specific primer on junction to performe quantitative PCR assay on the different transcript isoforms.
The same can be considered for the other exons.
This study to ABCA7 alternative variants is also very limited because it only focuse on the first 7 exons whereas ABCA7 contains 47 exons. I suppose the authors focus on these 7 first exons because they match with regions containing the highest frequence of CpG in introns : in fact these intronic CpG-rich introns could also be potential site of transcriptional start since at least from database, exon2 and exon7 can be alternative first exons. Note that within the 47 exons there are other CpG-rich introns.
Figure7, the variation of exon levels should be tested for their significant changes (statistically tested) if the authors consider these variations were reproducible.
3.2.5 / intron retention or other RNAprocessing events ?
In fact the apparent intron 6 retention observed by qPCR with primers targeting the exon6-intron6 boundary can be also a manifestation the alternative termination on exon6 leading to a long 3'UTR (NM_019112 / the coding transcript uc010dsa.3 (hg19 chr19:1,040,102-1,042,743 Size: 2,642 Total Exon Count: 6 Strand: +) or also see the non-coding transcript ENST00000526885). In other word further analysis are required to discrimate this RNA event between intron retention and alternative termination. For instance, the measure of the intron6-exon7 boundary, that in case of intron retention should be at the same level than the exon6-intron6 PCR. If not the alternative long 3'UTR should be considered . The problem may also com from the ENST00000435683 (uc010dsb.1) that starts from an alternative promoter intron6, creating a long 5'UTR joining to the exon7, so the PCR intron6-exon7 may also come from this alternative promoter.
As indicated line 644, if "the lower abundance of the exon 6–7 boundary is linked to higher intron 6 retention", it is maybe because there is a termination at exon6 (transcript uc010dsa.3), or because there is a regulation beteween the promoter/exon1 and the alternative promoter/exon7 giving variant ABCA7 protein in N-terminus domain.
Intron retention event are in general considered when this intron is found in cytoplasmic mRNA between the 5'exon and the 3'exon, i.e. when the intron is kept between two exons in mRNA fully processed and exported from the nuclei to be available for traduction. In total RNA extract this "intron retention" may be just an ongoing splicing intermediary with a longer life-time. RT-qPCR detecting these RNA events on biochemical fractionated cells (cytoplasmic extract/nuclei extract with good control : ie detecting the absence of U6 snRNA in cytoplasmic RNA for instance ), will prove the intron 6 is "retained". The authors should indicate whether this potential intron6 retention conserve an open reading frame or if a STOP codon occurs to give potentially a shortest protein.
It is particularly important to decipher the different transcriptional possibilities, as the lack of correlation between DNA methylation and the transcriptional event may be due to a messy interpretation of RNA levels.
exon1-exon2 junctions is described in database to be variant.
3.3.4 Proteomic study .
Again, because the analysis of protein by Mass-spectrometry in cytoplamic or nuclear fraction did not revealed ABCA7 peptides, the authors focused their study on other ABC transporter (line 705). This appears really unfocused information and should be the subject of an independent story.
The variation of ABCA7 proteins could be conducted by western blot : apparently different antibodies are commercially available and may allow to observed N-terminus truncated proteins depending of epitopes.
The variation of DNMTs should be linked also with TET enzymes but also Methyl-binding domain factors also important for global level of DNA methylation or its epigenetic interpretation by the transcriptional machinery.
Could it be possible to indicate splicing variation of the most abundant protein analysed by mass spect. Indeed it is also known that splicing variant of splicing factors (but also of MBDs or DNMT/TET enzymes) may have different functions. Could it be possible that this kind of splicing variation correlate to chemotherapy ?
As final remarks I'd like emphasize that even Gil Ast (ref 37) defends the idea that local DNA methylation can influence the alternative splicing decision, this clearly not the first regulator of the alternative splicing decisions. The main splicing regulators are of course the splicing factors, and it is understanded that in a context of "constant" splicing factor pool, the influence of DNA methylation on alternative splicing decision can be revealed. So it is very difficult by comparing different cell-lines with different expression level of splicing factors to correlate DNA methylation and splicing decsion. And this doe not mean that DNA methylation and alternative splicing decision are independent. Moreover, splicing regulation were in this field studied on alternative exons or alternative splice junction and not on exon abundantly included (see the selection of exons in . The status of "variant" exons in the ABCA7 first 7 exons is not really demonstrated. Showing in what extent the 7 first exons can be variable, will provide argument to indicate they could be influenced by DNA methylation level. Another point is that changes of alternative promoter that can be regulated by DNA methylation on CpG rich regions, may program different alternative splicing decision at distance, since this can influence the speed of the RNA polymerase II and the recruitment of splicing factors (see the Kornblihtt review PMID: 32857854). Overall, DNA methylation may directly or indirectly influence the alternative splicing decision in several ways and correlative studies may have great difficulty in bringing clear conclusions
other details :
Figures S1 and S2 were absent: I cannot judge their relevance.
line 249 : RNA integrity and quality were checked using the Qubit RNA IQ Assay Kit (Thermo Scientific, Austria).
It would be interesting to indicate in what sort of range the quality of RNA were considered good to further analysis since they were never shown in the paper in a classical way (RNA agarose gel or other).
Figure 7 : The levels of ABCA7 exons were measured by qPCR and normalized by the PUM1 housekeeping gene to compare the levels among the cell lines. It would be interesting to show the level of another housekeeping gene to demonstrate the PUM1 is well able to normalize the level of genes. In absence of this demonstration is these cell-lines, the variation of ABCA7 can be also a consequence of potential PUM1 variation.
Figure 8.
Why exon1-exon2 were not present in these comparison ?
Round 2
Reviewer 1 Report
The authors have addressed majority of suggestions. However, basal expression of ABCA7 in breast cancer cells is not clear, that could be due to technical issues.
Reviewer 3 Report
The study has been improved by the new data (Figure 9), but its interpretation has not been properly reported in the text. Formally, this should not change the conclusion about the lack of correlation between DNA methylation and the RNA event at the exon6 - intron6 boundary, but it is not just an "intron retention" as the authors call it. This single RT-qPCR primer pair measures different RNA events (as the new data prove) which may vary from cell to cell. The authors should explain this major limitation of their study or provide more complete data to support their claim. In its current form, the limited data do not support the "intron retention" conclusion.
In details :
The new PCR with the primers exon6-forward (F)+exon7-reverse (R) allowed the detection of the unspliced forms (=retention of intron6) (Figure 9d). It is noteworthy that the non-cancerous MCF10A cell line does not show this unspliced form (as well as intron5), in contrast to SK-BR3 or MDA-MB453, whereas all other tested introns (3, 4, 7) were detectable in MCF10A.
As RT-PCR efficiencies may vary, I suggest that the authors provide the ratio between the spliced and unspliced forms underneath the gels. This calculation will also show that the apparent level of intron6 retention, as measured by the ex6-int6 boundary (Fig. 8), is not consistent with the unspliced/spliced (ex6-int6-ex7 / ex6-ex7) ratio.
Comparing the results presented in this Fig9 with the data presented in Fig8, the results are apparently inconsistent: Fig8e shows less intron3,4,5 and 7 and more intron6 for MCF10A. This difference for MCF10A could be explained by the fact that the ex6-int6 boundary in fig8e does not only measure the retention of intron6 between exon6 and exon7, as shown by the absence of the unspliced ex6-int6-ex7 in fig9d. As a control, the observation of intron5 retention weakness (Fig. 9c) is very coherent with the level of ex5-int5 boundary between MCF10A (difference between ex5-ex6 and ex5-int5, DCt=5) and SKBR3 or MB453 (difference between ex5-ex6 and ex5-int5, DCt=3) (Fig. 8cde). The "incoherence" for intron6 can be interpreted by the use of an alternative transcription 3'-end in intron6 in MCF10A.
I'm sorry to insist, see my previous comment on quantifying the ex6-int6 and int6-ex7 boundaries to decipher between the intron retention of the ABCA7-202 transcript and the alternative end of the ABCA7-208 transcript, Fig 7. The intron retention formally involved both boundaries on two sides, not just one: otherwise it would be different transcription events (alternative end or alternative start). The quantification of intron6 retention should be performed taking into account the exon-intron boundaries on both sides: ex6-int6 and int6-ex7. The difference between the two sides will provide the quantification of the other RNA event, i.e. the alternative 3'end. (Note that an increase in the int6-ex7 boundary relative to the others suggests an alternative promoter in intron6 - see the transcript ENST0000435683 not shown in Figure 7 but described in the Ensembl database.). This MCF10A-specific termination in intron6, leading to the increase of transcript-208 (fig 7), would affect the synthesis of the intron6-exon7 boundary. This is also consistent with the strong correlation revealed by the author (lines 650-651). However, this correlation may have not the same cause-effect relation in all cell types.
The DNA methylation status at exon6/intron6 of MCF10A and of MDA-MB453 (cell lines used in fig 9) were not provided (Supl Fig S3 page 5). The absence of correlation between intron retention and DNA methylation will be much more convincing by using the comparable cell lines.
In conclusion, I only agree partially with the author conclusion (lines 912-914), and line 981 for treatment by drugs because what the authors named ‘intron retention” can be also something else : in the case of MCF10A it is an alternative end in intron6 and not an intron retention between ex6 and e7. In all other cell lines, in fact, the level of exon6-intron6 boundary can be a mixture at different ratio of the true intron retention and the transcript terminating in intron6. In these conditions, the lack of correlation can be explained.
The authors should explain clearly this strong limitation of their study, or alternatively provide a new data point with qPCR on the intron6-exon7 boundary (but can also be confused with an alternative promoter), or alternatively (and much more convincingly) a RT-PCR as in Figure 9 for all cell lines studied.
Minor point
Figure S3 : double labelling of different suplementary figures :
- page 5 of 6 - > refers to line 658
- page 6 of 6 - > refers to line 676 => should be labelled Figure S4
